# Learning to Plan in High Dimensions via Neural Exploration-Exploitation Trees

*Binghong Chen[1], *Bo Dai[2], Qinjie Lin[3], Guo Ye[3], Han Liu[3], Le Song[1,4]
[1]Georgia Institute of Technology    [2]Google Research, Brain Team
[3]Northwestern University    [4]Ant Financial

## Abstract

We propose a meta path planning algorithm named *Neural Exploration-Exploitation Trees (NEXT)* for learning from prior experience for solving new path planning problems in high dimensional continuous state and action spaces. Compared to more classical sampling-based methods like RRT, our approach achieves much better sample efficiency in high-dimensions and can benefit from prior experience of planning in similar environments. More specifically, NEXT exploits a novel neural architecture which can learn promising search directions from problem structures. The learned prior is then integrated into a UCB-type algorithm to achieve an online balance between *exploration* and *exploitation* when solving a new problem. We conduct thorough experiments to show that NEXT accomplishes new planning problems with more compact search trees and significantly outperforms state-of-the-art methods on several benchmarks.

## 1 Introduction

Path planning is a fundamental problem with many real-world applications, such as robot manipulation and autonomous driving. A simple planning problem within low-dimensional state space can be solved by first discretizing the continuous state space into a grid, and then searching for a path on top of it using graph search algorithms such as $A^*$ (Hart et al., 1968). However, due to the curse of dimensionality, these approaches do not scale well with the number of dimensions of the state space. For *high-dimensional* planning problems, people often resort to sampling-based approaches to avoid explicit discretization. Sampling-based planning algorithms, such as probabilistic roadmaps (PRM) (Kavraki et al., 1996), rapidly-exploring random trees (RRT) (LaValle, 1998), and their variants (Karaman & Frazzoli, 2011) incrementally build an implicit representation of the state space using probing samples. These generic algorithms typically employ a uniform sampler which does not make use of the structures of the problem. Therefore they may require lots of samples to obtain a feasible solution for complicated problems. To improve the sample efficiency, heuristic biased samplers, such as Gaussian sampler (Boor et al., 1999), bridge test (Hsu et al., 2003) and reachability-guided sampler (Shkolnik et al., 2009) have been proposed. All these sampling heuristics are designed manually to address specific structural properties, which may or may not be valid for a new problem, and may lead to even worse performance compared to the uniform proposal.

Online adaptation in path planning has also been investigated for improving sample efficiency in current planning problem. Specifically, Hsu et al. (2005) exploits online algorithms to dynamically adapts the mixture weights of several manually designed biased samplers. Burns & Brock (2005a;b) fit a model for the planning environment incrementally and use the model for planning. Yee et al. (2016) mimics the Monte-Carlos tree search (MCTS) for problems with continuous state and action spaces. These algorithms treat each planning problem *independently*, and the collected data from previous experiences and built model will be simply discarded when solving a new problem. However, in practice, similar planning problems may be solved again and again, where the problems are different but sharing common structures. For instance, grabbing a coffee cup on a table at different time are different problems, since the layout of paper and pens, the position and orientation of coffee cups may be different every time; however, all these problems show common structures of handling similar objects which are placed in similar fashions. Intuitively, if the common characteristics across problems can be learned via some shared latent representation, a planner based on such representation can then be transferred to new problems with improved sample efficiency.

---

*indicates equal contribution.

Several methods have been proposed recently to learn from past planning experiences to conduct more efficient and generalizable planning for future problems. These works are limited in one way or the other. Zucker et al. (2008); Zhang et al. (2018); Huh & Lee (2018) treat the sampler in the sampling-based planner as a stochastic policy to be learned and apply policy gradient or TD-algorithm to improve the policy. Finney et al. (2007); Bowen & Alterovitz (2014); Ye & Alterovitz (2017); Ichter et al. (2018); Kim et al. (2018); Kuo et al. (2018) apply imitation learning based on the collected demonstrations to bias for better sampler via variants of probabilistic models, *e.g.*, (mixture of) Gaussians, conditional VAE, GAN, HMM and RNN. However, many of these approaches either rely on specially designed local features or assume the problems are indexed by special parameters, which limits the generalization ability. Deep representation learning provides a promising direction to extract the common structure among the planning problems, and thus mitigate such limitation on hand-designed features. However, existing work, *e.g.*, motion planning networks (Qureshi et al., 2019), value iteration networks (VIN) (Tamar et al., 2016), and gated path planning networks (GPPN) (Lee et al., 2018), either apply off-the-shelf MLP architecture ignoring special structures in planning problems or can only deal with discrete state and action spaces in low-dimensional settings.

In this paper, we present *Neural EXploration-EXploitation Tree (NEXT)*, a **meta neural path planning** algorithm for high-dimensional continuous state space problems. The core contribution is a novel **attention-based neural architecture** that is capable of learning generalizable problem structures from previous experiences and produce promising search directions with automatic online **exploration-exploitation** balance adaption. Compared to existing learning-based planners,

- **NEXT is more generic**. We propose an architecture that can embed high dimensional continuous state spaces into low dimensional discrete spaces, on which a neural planning module is used to extract planning representation. These module will be learned end-to-end.
- **NEXT balances exploration-exploitation trade-off**. We integrate the learned neural prior into an upper confidence bound (UCB) style algorithm to achieve an online balance between exploration and exploitation when solving a new problem.

Empirically, we show that NEXT can exploit past experiences to reduce the number of required samples drastically for solving new planning problems, and significantly outperforms previous state-of-the-arts on several benchmark tasks.

## 1.1 RELATED WORKS

Designing non-uniform sampling strategies for random search to improve the planning efficiency has been considered as we discussed above. Besides the mentioned algorithms, there are other works along this line, including informed RRT* (Gammell et al., 2014) and batch informed Trees (BIT*) (Gammell et al., 2015) as the representative work. Randomized $A^*$ (Diankov & Kuffner, 2007) and sampling-based $A^*$ (Persson & Sharf, 2014) expand the search tree with hand-designed heuristics. These methods incorporate the human prior knowledge via hard-coded rules, which is fixed and unable to adapt to problems, and thus, may not universally applicable. Choudhury et al. (2018); Song et al. (2018) attempt to learn search heuristics. However, both methods are restricted to planning on discrete domains. Meanwhile, the latter one always employs an unnecessary hierarchical structure for path planning, which leads to inferior sample efficiency and extra computation.

The online exploration-exploitation trade-off is also an important issue in planning. For instance, Rickert et al. (2009) constructs a potential field sampler and tuned the sampler variance based on collision rate for the trade-off heuristically. Paxton et al. (2017) separates the action space into high-level discrete options and low-level continuous actions, and only considered the trade-off at the discrete option level, ignoring the exploration-exploitation in the fine action space. These existing works address the trade-off in an ad-hoc way, which may be inferior for the balance.

There have been sevearl non-learning-based planning methods that can also leverage experiences (Kavraki et al., 1996; Phillips et al., 2012) by utilizing search graphs created in previous problems. However, they are designed for largely fixed obstacles and cannot be generalized to unseen tasks from the same planning problems distribution.

## 2 SETTINGS FOR LEARNING TO PLAN

Let $\mathcal{S} \subseteq \mathbb{R}^q$ be the state space of the problem, *e.g.*, all the configurations of a robot and its base location in the workspace, $\mathcal{S}_{obs} \subsetneq \mathcal{S}$ be the obstacles set, $\mathcal{S}_{free} := \mathcal{S} \setminus \mathcal{S}_{obs}$ be the free space, $s_{init} \in \mathcal{S}_{free}$ be the initial state and $\mathcal{S}_{goal} \subsetneq \mathcal{S}_{free}$ be the goal region. Then the space of all collision-free paths can be defined as a continuous function $\Xi := \{\xi(\cdot) : [0,1] \to \mathcal{S}_{free}\}$. Let $c(\cdot) : \Xi \mapsto \mathbb{R}$

be the cost functional over a path. The optimal path planning problem is to find the optimal path in terms of cost $c(\cdot)$ from start $s_{init}$ to goal $\mathcal{S}_{goal}$ in free space $\mathcal{S}_{free}$, *i.e.*,

$$\xi^* = \mathrm{argmin}_{\xi \in \Xi} \ c(\xi), \quad \text{s.t. } \xi(0) = s_{init}, \ \xi(1) \in \mathcal{S}_{goal}. \tag{1}$$

Traditionally (Karaman & Frazzoli, 2011), the planner has direct access to $(s_{init}, \mathcal{S}_{goal}, c(\cdot))$ and the workspace map (Ichter et al., 2018; Tamar et al., 2016; Lee et al., 2018), $\mathtt{map}(\cdot) : \mathbb{R}^2$ or $\mathbb{R}^3 \mapsto \{0, 1\}$, (0: free spaces and 1: obstacles). Since $\mathcal{S}_{free}$ often has a very irregular geometry (illustrated in Figure 10 in Appendix A), it is usually represented via a *collision detection module* which is able to detect the obstacles in a path segment. For the same reason, the feasible paths in $\Xi$ are hard to be described in parametric forms, and thus, the nonparametric $\xi$, such as a sequence of interconnected path segments $[s_0, s_1], [s_1, s_2], \ldots, [s_{T-1}, s_T] \subset \mathcal{S}$ with $\xi(0) = s_0 = s_{init}$ and $\xi(1) = s_T$, is used with an additive cost $\sum_{i=1}^{T} c([s_{i-1}, s_i])$.

Assuming given the planning problems $\{U_i := (s_{init}, \mathcal{S}_{goal}, \mathcal{S}, \mathcal{S}_{free}, \mathtt{map}, c(\cdot))\}_{i=1}^{N}$ sampled from some distribution $\mathcal{U}$, we are interested in learning an algorithm $\mathtt{alg}(\cdot)$, which can produce the (nearly)-optimal path efficiently from the observed planning problems. Formally, the *learning to plan* is defined as

$$\mathtt{alg}^*(\cdot) = \mathrm{argmin}_{\mathtt{alg} \in \mathcal{A}} \ \mathbb{E}_{U \in \mathcal{U}} \left[ \ell(\mathtt{alg}(U)) \right], \tag{2}$$

where $\mathcal{A}$ denotes the planning algorithm family, and $\ell(\cdot)$ denotes some loss function which evaluates the quality of the generated path and the efficiency of the $\mathtt{alg}(\cdot)$, *e.g.*, size of the search tree. We elaborate each component in Eq (2) in the following sections. We first introduce the tree-based sampling algorithm template in Section 3, upon which we instantiate the $\mathtt{alg}(\cdot)$ via a novel attention-based neural parametrization in Section 4.2 with exploration-exploitation balance mechanism in Section 4.1. We design the $\ell$-loss function and the meta learning algorithm in Section 4.3. Composing every component together, we obtain the neural exploration-exploitation trees (NEXT) which achieves outstanding performances in Section 5.

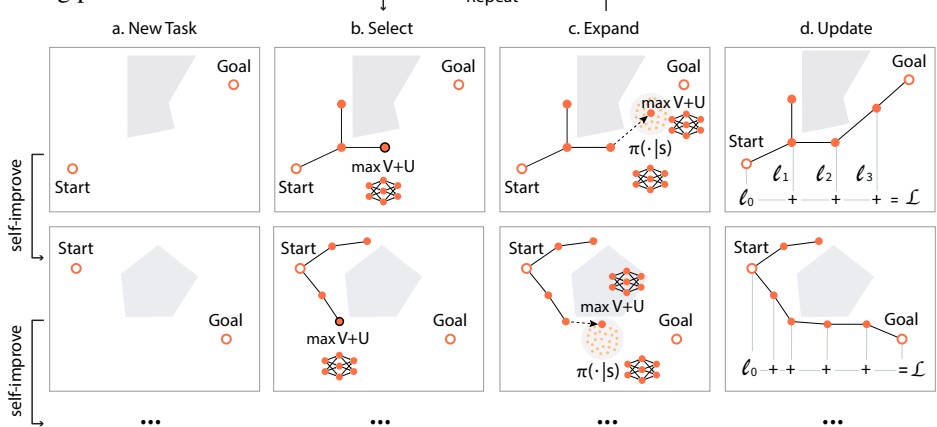

Figure 1: Illustration of NEXT. In each epoch, NEXT is executed on a randomly generated planning problem. The search tree grows with $\tilde{V}^*$ and $\tilde{\pi}^*$ guidance. $\{\tilde{V}^*, \tilde{\pi}^*\}$ will be updated according to the successful path. Such planning and learning iteration is continued interactively.

## 3 PRELIMINARIES

The sampling-based planners are more practical and become dominant for high-dimensional path planning problems (Elbanhawi & Simic, 2014). We describe a unifying view for many existing tree-based sampling algorithms (TSA), which we will also base our algorithm upon. More specifically, this family of algorithms maintain a search tree $\mathcal{T}$ rooted at the initial point $s_{init}$ and connecting all sampled points $\mathcal{V}$ in the configuration space with edge set $\mathcal{E}$. The tree will be expanded by incorporating more sampled states until some leaf reaches $\mathcal{S}_{goal}$. Then, a feasible solution for the path planning

---

**Algorithm 1:** Tree-based Sampling Algorithm

1 **Problem:**
$U = (s_{init}, \mathcal{S}_{goal}, \mathcal{S}, \mathcal{S}_{free}, \mathtt{map}, c(\cdot))$;
2 Initialize $\mathcal{T} = (\mathcal{V}, \mathcal{E})$ with $\mathcal{V} \leftarrow \{s_{init}\}$, $\mathcal{E} \leftarrow \emptyset$;
3 **for** $t \leftarrow 0$ **to** $T$ **do**
4     $s_{parent}, s_{new} \leftarrow \mathtt{Expand}(\mathcal{T}, U)$ ;
5     **if** $\mathtt{ObstacleFree}(s_{parent}, s_{new})$ **then**
6         $\mathcal{V} \leftarrow \mathcal{V} \cup \{s_{new}\}$ and $\mathcal{E} \leftarrow \mathcal{E} \cup \{[s_{parent}, s_{new}]\}$;
7         $\mathcal{T} \leftarrow \mathtt{Postprocess}(\mathcal{T}, U)$ ;
        ▷ Optional
8         **if** $s_{new} \in \mathcal{S}_{goal}$ **then**
9             **return** $\mathcal{T}$;

problem will be extracted based on the tree $\mathcal{T}$. The template of tree-based sampling algorithms is summarized in Algorithm 1 and illustrated in Fig. 1(c). A key component of the algorithm is the `Expand` operator, which generates the next exploration point $s_{new}$ and its parent $s_{parent} \in \mathcal{V}$. To ensure the feasibility of the solution, the $s_{new}$ must be **reachable** from $\mathcal{T}$, *i.e.*, $[s_{parent}, s_{new}]$ is collision-free, which is checked by a collision detection function. As we will discuss in Appendix B, by instantiating different `Expand` operators, we will arrive at many existing algorithms, such as RRT (LaValle, 1998) and EST (Hsu et al., 1997; Phillips et al., 2004).

One major limitation of existing TSAs is that they solve each problem independently from scratch and ignore past planning experiences in similar environments. We introduce the neural components into TSA template to form the learnable planning algorithm family $\mathcal{A}$, which can explicitly take advantages of the past successful experiences to bias the `Expand` towards more promising regions.

## 4 NEURAL EXPLORATION-EXPLOITATION TREES

Based on the TSA framework, we introduce a *learnable neural based* `Expand` operator, which can balance between exploration and exploitation, to instantiate $\mathcal{A}$ in Eq (2). With the self-improving training, we obtain the meta NEXT algorithm illustrated in Figure 1.

### 4.1 GUIDED PROGRESSIVE EXPANSION

We start with our design of the `Expand`. We assume having an estimated value function $\tilde{V}^*(s|U)$, which stands for the optimal cost from $s$ to target in planning problem $U$, and a policy $\tilde{\pi}^*(s'|s, U)$, which generates the promising action $s'$ from state $s$. The concrete parametrization of $\tilde{V}^*$ and $\tilde{\pi}^*$ will be explained in Section 4.2 and learned in Section 4.3. We will use these functions to construct the learnable `Expand` with explicit exploration-exploitation balancing.

---

**Algorithm 2:** NEXT :: Expand$(\mathcal{T} = (\mathcal{V}, \mathcal{E}), U)$

---

1  $s_{parent} \leftarrow \text{argmax}_{s \in \mathcal{V}} \, \phi(s)$ ;  $\quad\quad\quad\quad \triangleright$ Selection
2  $\{s_1, \ldots, s_k\} \overset{iid.}{\sim} \tilde{\pi}^*(s'|s_{parent}, U)$ ;  $\triangleright$ Candidates
3  $s_{new} \leftarrow \text{argmax}_{s' \in \{s_1, \ldots, s_k\}} \, \phi(s')$;  $\quad \triangleright$ Expansion
4  **return** $s_{parent}, s_{new}$;

---

The `Expand` operator will expand the current search tree $\mathcal{T}$ by a new neighboring state $s_{new}$ around $\mathcal{T}$. We design the expansion as a two-step procedure: **(i)** select a state $s_{parent}$ from existing tree $\mathcal{T}$; **(ii)** expand a state $s_{new}$ in the neighborhood of $s_{parent}$. More specifically,

**Selecting $s_{parent}$ from $\mathcal{T}$ in step (i).** Consider the negative value function $-\tilde{V}^*(s|U)$ as the rewards $r(s)$, step **(i)** shares some similarity with the multi-armed bandit problem by viewing existing nodes $s \in \mathcal{V}$ as arms. However, the vanilla UCB algorithm is not directly applicable, since the number of states is increasing as the algorithm proceeds and the value of these adjacent states are naturally correlated. We address this challenge by modeling the correlation explicitly as in contextual bandits. Specifically, we parametrize the UCB of the reward function as $\phi(s)$, and select a node from $\mathcal{T}$ according to $\phi(s)$

$$ s_{parent} = \text{argmax}_{s \in \mathcal{V}} \quad \phi(s) := \bar{r}_t(s) + \lambda \sigma_t(s), \tag{3} $$

where $\bar{r}_t$ and $\sigma_t$ denote the average reward and variance estimator after $t$-calls to `Expand`. Denote the sequence of $t$ selected tree nodes so far as $\mathcal{S}_t = \{s^1_{parent}, \ldots, s^t_{parent}\}$, then we can use kernel smoothing estimator for $\bar{r}_t(s) = \frac{\sum_{s' \in \mathcal{S}_t} k(s', s) r(s')}{\sum_{s' \in \mathcal{S}_t} k(s', s)}$ and $\sigma_t(s) = \sqrt{\frac{\log \sum_{s' \in \mathcal{S}_t} w(s')}{w(s)}}$ where $k(s', s)$ is a kernel function and $w(s) = \sum_{s' \in \mathcal{S}_t} k(s', s)$. Other parametrizations of $\bar{r}_t$ and $\sigma_t$ are also possible, such as Gaussian Process parametrization in Appendix C. The average reward exploits more promising states, while the variance promotes exploration towards less frequently visited states; and the exploration versus exploitation is balanced by a tunable weight $\lambda > 0$.

**Expanding a reachable $s_{new}$ in step (ii).** Given the selected $s_{parent}$, we consider expanding a reachable state in the neighborhood $s_{parent}$ as an *infinite-armed* bandit problem. Although one can first samples $k$ arms uniformly from a neighborhood around $s_{parent}$ and runs a UCB algorithm on the randomly generated finite arms (Wang et al., 2009), such uniform sampler ignores problem structures, and will lead to unnecessary samples. Instead we will employ a policy $\tilde{\pi}^*(s'|s, U)$[1] for guidance when generating the candidates. The final choice for next move will be selected from these candidates with $\max \phi(s)$ defined in (3). As explained in more details in Section 4.2, $\tilde{\pi}^*$ will be

---

[1]In the path planning setting, we use $\tilde{\pi}^*(s'|s, U)$ and $\tilde{\pi}^*(a|s, U)$ interchangeably as the action is next state.

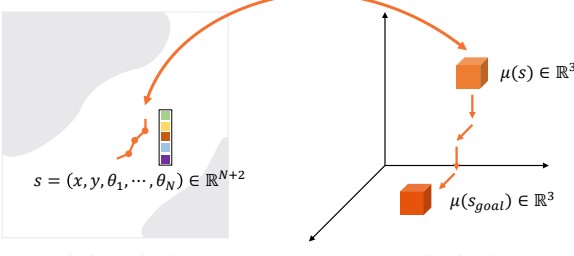

Figure 2: Our neural network model maps a $N$-link robot from the original planning space (a $(N + 2)$-d configuration space) to a 3d discrete latent planning space in which we plan a path using value iteration. The result of value iteration is then used as features for defining $\tilde{V}^*(s|U)$ and $\tilde{\pi}^*(s'|s, U)$.

Original Planning Space      Latent Planning Space

trained to mimic previous successful planning experiences across different problems, that is, biasing the sampling towards the states with higher successful probability.

With these detailed step **(i)** and **(ii)**, we obtain `NEXT :: Expand` in Algorithm 2 (illustrated in Figure 1(b) and (c)). Plugging it into the TSA in 1, we construct $\text{alg}(\cdot) \in \mathcal{A}$ which will be learned.

The guided progressive expansion bears similarity to MCTS but deals with high dimensional continuous spaces. Moreover, the essential difference lies in the way to select state in $\mathcal{T}$ for expansion: the MCTS only expands the leaf states in $\mathcal{T}$ due to the hierarchical assumption, limiting the exploration ability and incurring extra unnecessary UCB sampling for internal traversal in $\mathcal{T}$; while the proposed operation enables expansion from each visited state, particularly suitable for path planning problems.

## 4.2 NEURAL ARCHITECTURE FOR VALUE FUNCTION AND EXPANSION POLICY

In this section, we will introduce our neural architectures for $\tilde{V}^*(s|U)$ and $\tilde{\pi}^*(s'|s, U)$ used in `NEXT :: Expand`. The proposed neural architectures can be understood as first embedding the state and problem into a **discrete** latent representation via an attention-based module in Section 4.2.1, upon which the neuralized value iteration, introduced in Section 4.2.2, is performed to extract features for defining $\tilde{V}^*(s|U)$ and $\tilde{\pi}^*(s'|s, U)$, as illustrated in Figure 2.

### 4.2.1 CONFIGURATION SPACE EMBEDDING

Our network for embedding high-dimension configuration space into a latent representation is designed based on an attention mechanism. More specifically, let $s^w$ denote the workspace in state and $s^h$ denote the remaining dimensions of the state, *i.e.* $s = (s^w, s^h)$. $s^w$ and $s^h$ will be embedded by different sub-neural networks and combined for the final representation, as illustrated in Figure 3. For simplicity of exposition, we will focus on the 2d workspace, *i.e.*, $s^w \in \mathbb{R}^2$. However, our method applies to 3d workspace as well.

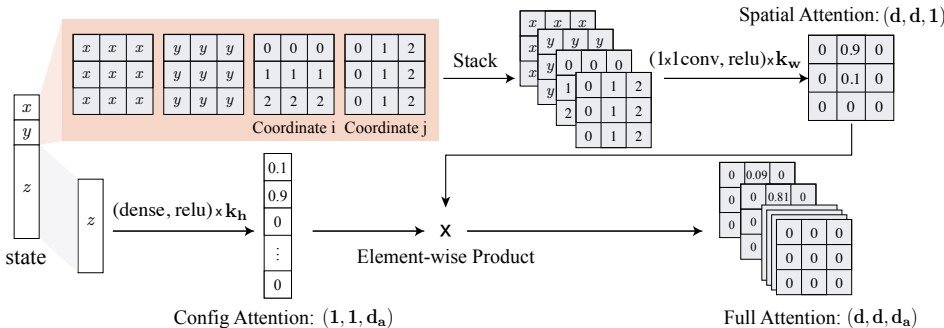

Figure 3: Attention-based state embedding module. $s^w = (x, y)$ and $\mathbf{z} = s^h$. The upper part is spatial attention, with the first two channels being $x$ and $y$, and the last two channels being constant templates with the row and column coordinates, as shown with a $d$ set to 3. The bottom module learns the representation for $\mathbf{z}$. The final embedding is obtained by outer-product of these two attention parts.

- **Spatial attention.** The workspace information $s^w$ will be embedded as $\mu^w(s^w) \in \mathbb{R}^{d \times d}$, $d$ is a hyperparameter related to `map` (see remark below). The spatial embedding module (upper part in Figure 3) is composed of $k_w$ convolution layers, *i.e.*,

$$\mu^w(s^w) = \text{softmax2d}(f_{k_w}^w(s^w)), \qquad f_{i+1}^w(s^w) = \text{relu}(\theta_i^w \oplus f_i^w(s^w)), \qquad (4)$$

where $\theta_i^w$ denotes the convolution kernels, $\oplus$ denotes the convolution operator and $f_i^w(s^w) \in \mathbb{R}^{d \times d \times d_i}$ with $d_i$ channels. The first layer $f_0^w$ is designed to represent $s^w$ into a $d \times d \times 4$ tensor as $f_0^w(s^w)_{ij} = [s_1^w, s_2^w, i, j]$, $i, j = 1, \ldots, d$, without any loss of information.

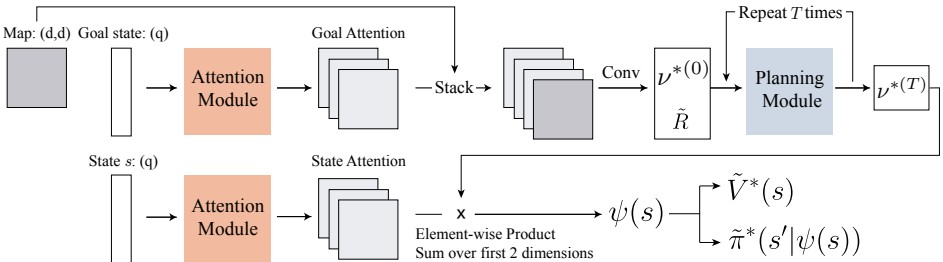

Figure 4: Overall model architecture. Current and goal states are embedded through attention module. Then the embedding of the goal state is concatenated with the map to produce $\nu^{*(0)}$ and $\tilde{R}$ as the input to the planning module. The output of the planning module is aggregated with the embedding of the current state to produce feature $\psi(s)$ for defining $\tilde{V}^*$ and $\tilde{\pi}^*$.

- **Configuration attention.** The remaining configuration state information will be embedded as $\mu^h(s^h)$ through $k_h$ fully-connected layers (bottom-left part in Figure 3), *i.e.*

$$\mu^h(s^h) = \texttt{softmax}(f_{k_h}^h(s^h)), \qquad f_{i+1}^h(s^h) = \texttt{relu}(\theta_i^h f_i^h(s^h) + b_i),$$

where $\mu^h(s^h) \in \mathbb{R}^{d_a}$ and $f_0^h(s^h) = s^h$.

The final representation $\mu_\theta(s)$ will be obtained by multiplying $\mu^w(s^w)$ with $\mu^h(s^h)$ element-wisely, $\mu_\theta(s)_{ijl} = \mu^w(s^w)_{ij} \cdot \mu^h(s^h)_l$, which is a $d \times d \times d_a$ tensor attention map with $\mu_\theta(s)_{ijl} \geqslant 0$, and $\sum_{ijl} \mu_\theta(s)_{ijl} = 1$ (bottom-right part in Figure 3). Intuitively, one can think of $d_a$ as the level of the *learned discretization* of the configuration space $s^h$, and the entries in $\mu$ softly assign the actual state $s$ to these discretized locations. $\theta := (\{\theta_i^w\}_{i=0}^{k_w-1}, \{\theta_i^h, b_i\}_{i=0}^{k_h-1})$ are the parameters to be learned.

**Remark (different map size):** To process map using convolution neural networks, we resize it to a $d \times d$ image, with the same size as the spatial attention in Eq (4), where $d$ is a hyperparameter.

### 4.2.2 NEURAL VALUE ITERATION

We then apply neuralized value iteration on top of the configuration space embedding to extract further planning features (Tamar et al., 2016; Lee et al., 2018). Specifically, we first produce the embedding $\mu_\theta(s_{goal})$ of the center of the goal region $s_{goal}$. We execute $T$ steps of neuralized Bellman updates (planning module in Figure 4) in the embedding space,

$$\nu^{*(t)} = \min\left(W_1 \oplus \left[\nu^{*(t-1)}, \tilde{R}\right]\right), \quad \text{with} \quad (\nu^{*(0)}, \tilde{R}) = \sigma\left(W_0 \oplus [\mu_\theta(s_{goal}), \texttt{map}]\right),$$

and obtain $\nu^{*(T)} \in \mathbb{R}^{d \times d \times d_a \times p}$. Both $W_0$ and $W_1$ are 3d convolution kernels, $\min$ implements the pooling across channels. Accordingly, $\nu^{*(T)}$ now can be understood as a latent representation of the value function $\tilde{V}^*(\cdot)$ in learned embedding space for the problem $U$ with $s_{goal}$ in map.

To define the value function for particular state $s$, *i.e.*, $\tilde{V}^*(s|U)$, from the latent representation $\nu^{*(T)}$, we first construct another attention model between the embedding of state $s$ using $\mu_\theta(s)$ and $\nu^{*(T)}$, *i.e.*, $\psi(s)_k = \sum_{ijl} \nu_{ijlk}^{*(T)} \cdot \mu_\theta(s)_{ijl}$, for $k = 1, \dots, p$. Finally we define

$$\tilde{V}^*(s|U) = h_{W_2}(\psi(s)), \quad \text{and} \quad \tilde{\pi}^*(s'|s, U) = \mathcal{N}(h_{W_3}(\psi(s)), \sigma^2) \qquad (5)$$

where $h_{W_2}$ and $h_{W_3}$ are fully connected dense layers with parameters $W_2$ and $W_3$ respectively, and $\mathcal{N}(h_{W_3}(\psi(s)), \sigma^2)$ is a Gaussian distribution with variance $\sigma^2$. Note that we also parametrize the policy $\tilde{\pi}^*(s'|s, U)$ using the embedding $\nu^{*(T)}$, since the policy is connected to the value function via $\pi^*(s'|s, U) = \operatorname{argmin}_{s' \in \mathcal{S}} c([s, s']) + V^*(s'|U)$. It should also be emphasized that in our parametrization, the calculation of $\nu^{*(T)}$ only relies on the $\mu_\theta(s_{goal})$, which can be reused for evaluating $\tilde{V}^*(s|U)$ and $\tilde{\pi}^*(s'|s, U)$ over different $s$, saving computational resources. Using this trick the algorithm runs $10\times$-$100\times$ faster empirically.

The overall model architecture in $\texttt{alg}(\cdot)$ is illustrated in Figure 4. The parameters $W = (W_0, W_1, W_2, W_3, \theta)$ will be learned together by our *meta self-improving learning*. For the details of the parameterization and the size of convolution kernels in our implementation, please refer to Figure 12 in Appendix D.

### 4.3 META SELF-IMPROVING LEARNING

The learning of the planner $\mathtt{alg}(\cdot)$ reduces to learning the parameters in $\tilde{V}^*(s|U)$ and $\tilde{\pi}^*(s'|s, U)$ and is carried out while planning experiences accumulate. We do not have an explicit training and testing phase separation. Particularly, we use a mixture of $\mathtt{RRT :: Expand}$ and $\mathtt{NEXT :: Expand}$ with probability $\epsilon$ and $1 - \epsilon$, respectively, inside the $\mathtt{TSA}$ framework in Algorithm 1. The RRT* postprocessing step is used in the template. The $\epsilon$ is set to be 1 initially since $\{\tilde{V}^*, \tilde{\pi}^*\}$ are not well-trained, and thus, the algorithm behaves like RRT*. As the training proceeds, we anneal $\epsilon$ gradually as the sampler becomes more efficient.

The dataset $\mathcal{D}_n = \{(\mathcal{T}_j, U_j)\}_{j=1}^n$ for the $n$-th training epoch is collected from the previous successful planning experiences across *multiple random problems*. We fix the size of dataset and update $\mathcal{D}$ in the same way as experience reply buffer (Lin, 1992; Schaul et al., 2015). For an experience $(\mathcal{T}, U) \in \mathcal{D}_n$, we can reconstruct the successful path $\{s^i\}_{i=1}^m$ from the search tree $\mathcal{T}$ ($m$ is the number of segments), and the value of each state $s^i$ in the path will be the sum of cost to the goal region, *i.e.*, $y^i := \sum_{l=i}^{m-1} c([s^l, s^{l+1}])$. We learn $\{\tilde{V}^*, \tilde{\pi}^*\}$ by optimizing objective

$$\min_W \sum_{(\mathcal{T}, U) \in \mathcal{D}_n} \ell(\tilde{V}^*, \tilde{\pi}^*; \mathcal{T}, U) := -\sum_{\mathcal{D}_n} \sum_{i=1}^{m-1} \log \tilde{\pi}^*\left(s^{i+1}|s^i\right) + \sum_{i=1}^m (\tilde{V}^*\left(s^i\right) - y^i)_2^2 + \lambda \|W\|^2 . \quad (6)$$

The loss (6) pushes the $\tilde{V}^*$ and $\tilde{\pi}^*$ to chase the successful trajectories, providing effective guidance in $\mathtt{alg}(\cdot)$ for searching, and therefore leading to efficient searching procedure with less sample complexity and better solution. On one hand, the value function and policy estimation $\{\tilde{V}^*, \tilde{\pi}^*\}$ is improved based upon the successful outcomes from NEXT itself on previous problems. On the other hand, the updated $\{\tilde{V}^*, \tilde{\pi}^*\}$ will be applied in the next epoch to improve the performance. Therefore, the training is named as *Meta Self-Improving Learning (MSIL)*. Since all the trajectories we collected for learning are feasible, the reachability of the proposed samples is enforced implicitly via imitating these successful paths.

By putting every components together into the learning to plan framework in Eq (2), the overall procedure is summarized in Algorithm 3 and illustrated in Figure 1.

---

**Algorithm 3:** Meta Self-Improving Learning

1   Initialize dataset $\mathcal{D}_0$;
2   **for** epoch $n \leftarrow 1$ **to** $N$ **do**
3      Sample a planning problem $U$;
4      $\mathcal{T} \leftarrow \mathtt{TSA}(U)$ with $\epsilon \sim \mathcal{U}nif[0, 1]$, and $\epsilon \cdot \mathtt{RRT :: Expand} + (1-\epsilon) \cdot \mathtt{NEXT :: Expand}$; Postprocessing with $\mathtt{RRT^* :: Postprocess}$;
5      $\mathcal{D}_n \leftarrow \mathcal{D}_{n-1} \cup \{(\mathcal{T}, U)\}$ if successful else $\mathcal{D}_{n-1}$;
6      **for** $j \leftarrow 0$ **to** $L$ **do**
7          Sample $(\mathcal{T}_j, U_j)$ from $\mathcal{D}_n$;
8          Reconstruct sub-optimal path $\{s^i\}_{i=1}^m$ and the cost of paths based on $\mathcal{T}_j$;
9          Update parameters $W \leftarrow W - \eta \nabla_W \ell(\tilde{V}^*, \tilde{\pi}^*; \mathcal{T}_j, U_j)$;
10     Anneal $\epsilon = \alpha \epsilon, \alpha \in (0, 1)$;

---

## 5 EXPERIMENTS

In this section, we evaluate the proposed NEXT empirically on different planning tasks in a variety of environments. Comparing to the existing planning algorithms, NEXT achieves the state-of-the-art performances, in terms of both success rate and the quality of the found solutions. We further demonstrate the power of the proposed two components by the corresponding ablation study. We also include a case study on a real-world robot arm control problem at the end of the section.

### 5.1 EXPERIMENT SETUP

**Benchmark environments.** We designed four benchmark tasks to demonstrate the effectiveness of our algorithm for high-dimensional planning. The first three involve planning in a 2d workspace with a 2 DoF (degrees of freedom) point robot, a 3 DoF stick robot and a 5 DoF snake robot, respectively. The last one involves planning a 7 DoF spacecraft in a 3d workspace. For all problems in each benchmark task, the workspace maps were randomly generated from a fixed distribution; the initial and goal states were sampled uniformly randomly in the free space; the cost function $c(\cdot)$ was set as the sum of the Euclidean path length and the control effort penalty of rotating the robot joints.

**Baselines.** We compared NEXT with RRT* (Karaman & Frazzoli, 2011), BIT* (Gammell et al., 2015), CVAE-plan (Ichter et al., 2018), Reinforce-plan (Zhang et al., 2018), and an improved

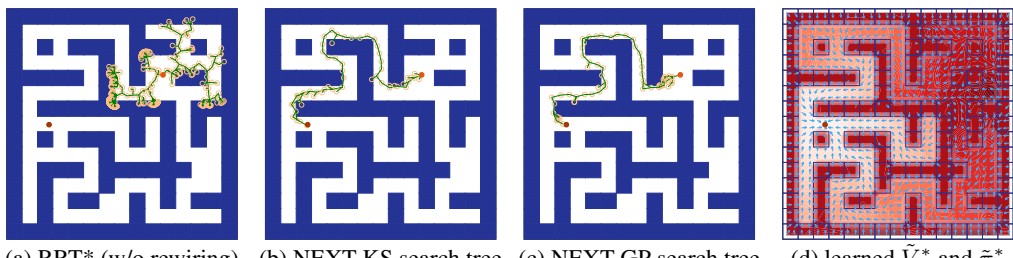

(a) RRT* (w/o rewiring)  (b) NEXT-KS search tree  (c) NEXT-GP search tree  (d) learned $\tilde{V}^*$ and $\tilde{\pi}^*$

Figure 5: Search trees and the learned $\tilde{V}^*$ and $\tilde{\pi}^*$ produced by NEXT. Obstacles are colored in blue. The start and goal locations are denoted by orange and brown dots. In (a) to (c), samples are represented with yellow circles. In (d), the level of redness denotes the value of the cost-to-go estimate $\tilde{V}^*$. The cyan arrows point from a given state $s$ to the mean of the learned policy $\tilde{\pi}^*(s'|s, U)$.

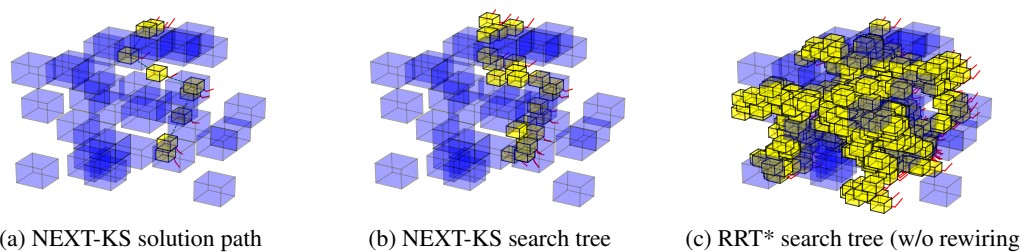

(a) NEXT-KS solution path  (b) NEXT-KS search tree  (c) RRT* search tree (w/o rewiring)

Figure 6: Search trees and a solution path produced in an instance of spacecraft planning. The 7 DOF spacecraft has a yellow body and two 2 DOF red arms. NEXT-KS produced a nearly minimum viable search tree while RRT* failed to find a path within limited trials.

version of GPPN (Lee et al., 2018) in terms of both planning time and solution quality. RRT* and BIT* are two widely used effective instances of TSA in Algorithm 1. In our experiments, we equipped RRT* with the goal biasing heuristic to improve its performance. BIT* adopts the informed search strategy (Gammell et al., 2015) to accelerate planning. CVAE-plan and Reinforce-plan are two learning-enhanced TSA planners proposed recently. CVAE-plan learns a conditional VAE as the sampler (Sohn et al., 2015), which will be trained by near-optimal paths produced by RRT*. Reinforce-plan learns to do rejection sampling with policy gradient methods. For the improved GPPN, we combined its architecture for map with a fully-connected MLP for the rest state, such that it can be applied to high-dimensional continuous spaces. Please refer to Appendix E for more details.

**Settings.** For each task, we randomly generated 3000 different problems from the same distribution without duplicated maps. We trained all learning-based baselines using the first 2000 problems, and reserved the rest for testing. The parameters for RRT* and BIT* are also tuned using the first 2000 problems. For NEXT, we let it improve itself using MSIL over the first 2000 problems. In this period, for every 200 problems, we updated its parameters and annealed $\epsilon$ once.

## 5.2 RESULTS AND ANALYSIS

**Comparison results.** Examples of all four environments are illustrated in Appendix F.1 and Figure 6, where NEXT finds high-quality solutions as shown. We also illustrated the comparison of the search trees on two 2d and 7d planning tasks between NEXT and RRT* in Figure 5 (a)-(c) and Figure 6 (b) and (c). Obviously, the proposed NEXT algorithm explores with guidance and achieves better quality solutions with fewer samples, while the RRT* expands randomly which may fail to find a solution. The learned $\tilde{V}^*$ and $\tilde{\pi}^*$ in the 2d task are also shown in Figure 5(d). As we can see, they are consistent with our expectation, towards the ultimate target in the map. For more search tree comparisons for all four experiments, please check Figure 18, 19, 20 and 21 in Appendix F.

To systematically evaluate the algorithms, we recorded the cost of time (measured by the number of collision checks used) to find a collision-free path, the success rate within time limits, and the cost of the solution path for each run. The results of the reserved 1000 test problems of each environment are shown in the top row of Figure 7. We set the maximal number of samples as 500 for all algorithms. Both the kernel smoothing (NEXT-KS) and the Gaussian process (NEXT-GP) version of NEXT achieves the state-of-the-art performances, under all three criteria in all test environments. Although the BIT* utilizes the heuristic particularly suitable for 3d maze in 7d task and performs quite well,

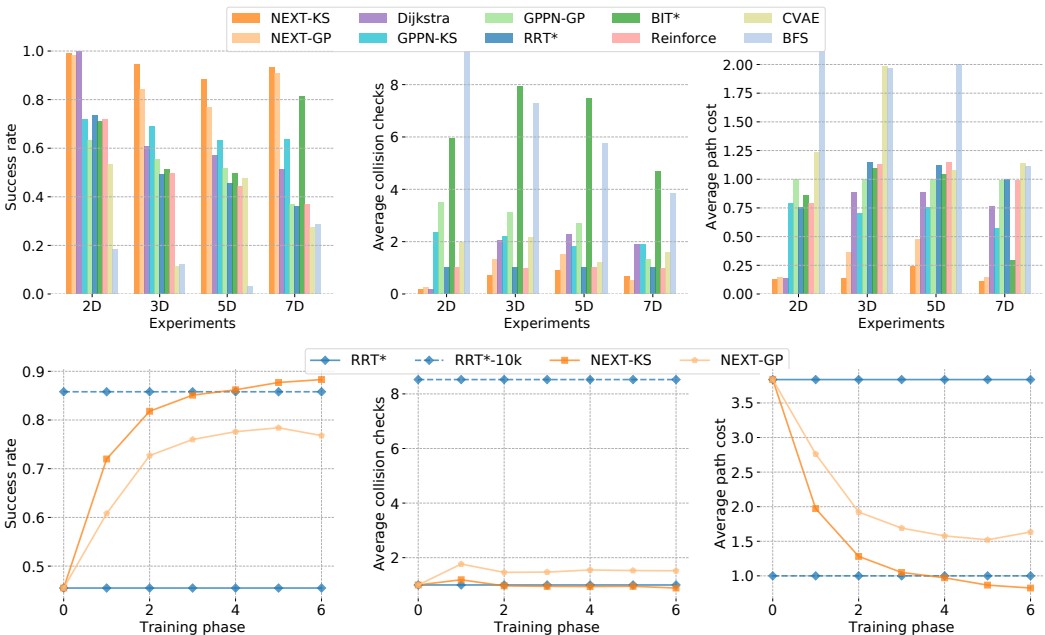

Figure 7: First row: histograms of results, in terms of success rate, average collision checks, and average cost of the solution paths; Second row: NEXT improvement curves in the 5d experiments. All algorithms are set to use up to 500 samples, except RRT*-10k, which uses 10,000 samples. The value of collision checks and path costs are normalized w.r.t. the performance of RRT*.

the NEXT algorithm still outperform every competitor by a large margin, no matter learning-based or prefixed heuristic planner, demonstrating the advantages of the proposed NEXT algorithm.

**Self-improving.** We plot the performance improvement curves of our algorithms on the 5d planning task in the bottom row of Figure 7. For comparison, we also plot the performance of RRT*. At the beginning phase of self-improving, our algorithms are comparable to RRT*. They then gradually learn from previous experiences and improve themselves as they see more problems and better solutions. In the end, NEXT-KS is able to match the performance of RRT*-10k using only one-twentieth of its samples, while the competitors perform consistently without any improvements.

Due to the space limits, we put improvement curves on other environments in Figure 17 and the quantitative evaluation in Table 1, 2, and 3 in Appendix F. Please refer to the details there.

## 5.3 Ablation Studies

**Ablation study I: guided progressive expansion.** To demonstrate the power of NEXT :: Expand, we replace it with breadth-first search (BFS) (Kim et al., 2018), another expanding strategy, while keeping other components the same. Specifically, BFS uses a search queue in planning. It repeatedly pops a state $s$ out from the search queue, samples $k$ states from $\pi(\cdot|s)$, and pushes all generated samples and state $s$ back to the queue, until the goal is reached. For fairness, we use the learned sampling policy $\pi(s'|s, U)$ by NEXT-KS in BFS. As shown in Figure 7, BFS obtained worse paths with a much larger number of collision checks and far lower success rate, which justifies the importance of the balance between exploration versus exploitation achieved by the proposed NEXT :: Expand.

**Ablation study II: neural architecture.** To further demonstrate the benefits of the proposed neural architecture for learning generalizable representations in high-dimension planning problems, We replaced our attention-based neural architecture with an improved GPPN, as explained in Appendix E, for ablation study. We extended the GPPN for continuous space by adding an extra reactive policy network to its final layers. We emphasize the original GPPN is not applicable to the tasks in our experiments. Intuitively, the improved GPPN first produces a 'rough plan' by processing the robot's discretized workspace positions. Then the reactive policy network predicts a continuous action from both the workspace feature and the full configuration state of the robot. We provide more preference to the improved GPPN by training it to imitate the near-optimal paths produced by RRT* in the training problems. During test time it is also combined with both versions of the guided progressive expansion operators. As we can see, both GPPN-KS and GPPN-GP are clearly much

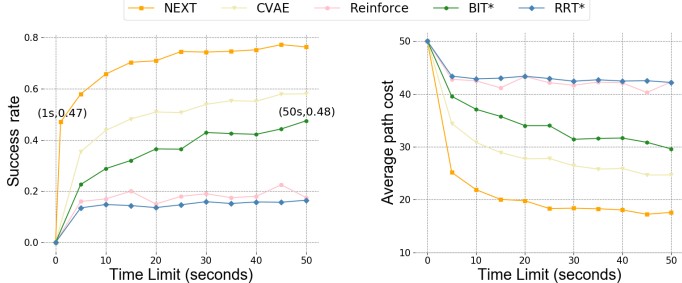

Figure 8: The success rate and average path cost of the different planners under varying time limits. Running NEXT for 1 second achieves the same success rate as running BIT* for 50 seconds.

worse than NEXT-KS and NEXT-GP, demonstrating the advantage of our proposed attention-based neural architecture in high-dimensional planning tasks.

**Ablation study III: learning versus heuristic.** The NEXT algorithm in Figure 5 shows similar behavior as the Dijkstra heuristic, *i.e.* sampling on the shortest path connecting the start and the goal in workspace. However, in higher dimensional space, the Dijkstra heuristic will fail. To demonstrate that, we replace the policy and value network with Dijkstra heuristic, using the best direction in workspace to guide sampling. NEXT performs much better than Dijkstra in all but the 2d case, in which the workspace happens to be the state space.

## 5.4 CASE STUDY: ROBOT ARM CONTROL

We conduct a real-world case study on controlling robot arms to move objects on a shelf. On this representative real-time task, we demonstrate the advantages of the NEXT in terms of the wall-clock.

In each planning task, there is a shelf of multiple levels, with each level horizontally divided into multiple bins. The task is to plan a path from a location in one bin to another, *i.e.*, the end effectors of the start and goal configurations are in different bins. The heights of levels, widths of bins, and the start and goal are randomly drawn from some fixed distribution. Different from previous experiments, the base of the robot is fixed. We consider the BIT* instead of RRT* as the imperfect expert in 3000 training problems. We then evaluate the algorithm on a separated 1000 testing problems. We compare NEXT(-KS) with the highly tuned BIT* and RRT* in OMPL, and also CVAE-plan and Reinforce-plan in Figure 8. As seen from the visualization of the found paths in Figure 9, this is a very difficult task. Our NEXT outperforms the baselines by a large margin, requiring only 1 second to reach the same success rate as running 50 seconds of BIT*.

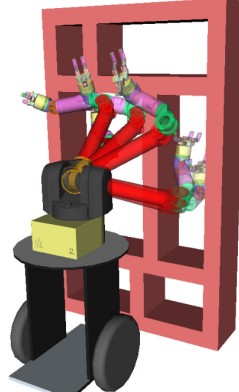

Figure 9: The collision-free path produced by NEXT for robot arm planning. The start and goal configurations have end-effectors in different bins of the shelf.

Due to space limits, we put details of the experiment setups, more results and analysis in Appendix F.4.

## 6 CONCLUSION

In this paper, we propose a self-improving planner, Neural EXploration-EXploitation Trees (NEXT), which can generalize and achieve better performance with experiences accumulated. The algorithm achieves a delicate balance between exploration-exploitation via our carefully designed UCB-type expansion operation. To obtain the generalizable ability across different problems, we proposed a new parametrization for the value function and policy, which captures the Bellman recursive structure in the high-dimensional continuous state and action space. We demonstrate the power of the proposed algorithm by outperforming previous state-of-the-art planners with significant margins on planning problems in a variety of different environments.

## ACKNOWLEDGEMENT

We thank the Google Research Brain team members for helpful thoughts and discussions as well as the anonymous reviewers for their insightful comments and suggestions. This work is supported in part by NSF grants CDS&E-1900017 D3SC, CCF-1836936 FMitF, IIS-1841351, CAREER IIS-1350983 to L.S, and by NSF grants BIGDATA 1840866, CAREER 1841569, TRIPODS 1740735, DARPA-PA-18-02-09-QED-RML-FP-003, an Alfred P Sloan Fellowship, a PECASE award to H.L.

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

# Appendix

## A ILLUSTRATION OF THE DIFFICULTY IN PLANNING PROBLEMS

The Figure 10(a) illustrates a concrete planning problem for a stick robot in 2d workspace. With one extra continuous action for rotation, the configuration state is visualized in Figure 10(b), which is highly irregular and unknown to the planner.

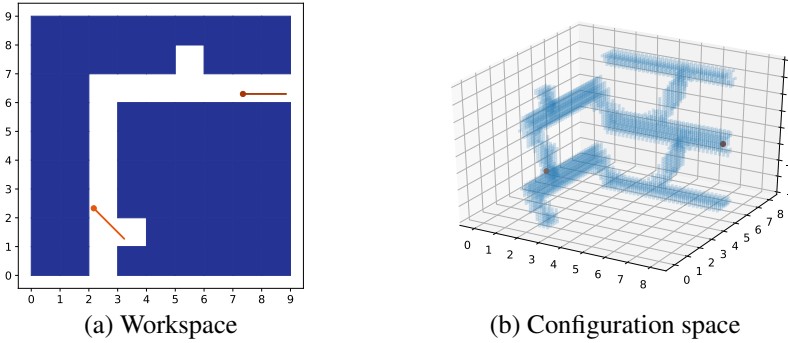

(a) Workspace        (b) Configuration space

Figure 10: Different views of the same planning problem. In (a) we color the obstacles, the starting and the goal position of the robot in deep blue, orange and brown, respectively. The stick robot can move and rotate. The corresponding configuration space is 3d, as visualized in (b), with the extra dimension being the rotation angle w.r.t. the x-axis. The blue region indicates the feasible state space, *i.e.*, the set of collision-free states. The starting and the goal position are denoted with an orange and a brown dot, respectively. Although the workspace looks trivial, the configuration space is irregular, which makes the planning difficult.

## B MORE PRELIMINARIES

**Tree-based sampling planner** The tree-based sampling planner algorithm is illustrated in Figure 11. The Expand in Algorithm 1 operator returns an existing node in the tree $s_{parent} \in \mathcal{V}$ and a new state $s_{new} \in \mathcal{S}$ sampled from the neighborhood of $s_{parent}$. Then the line segment $[s_{parent}, s_{new}]$ is passed to function ObstacleFree for collision checking. If the line segment $[s_{parent}, s_{new}]$ is collision-free (no obstacle in the middle, or called **reachable** from $\mathcal{T}$), then $s_{new}$ is added to the tree vertex set $\mathcal{V}$, and the line segment is added to the tree edge set $\mathcal{E}$. If the newly added node $s_{new}$ has reached the target $\mathcal{S}_{goal}$, the algorithm will return. Optionally, some concrete algorithms can define a Postprocess operator to refine the search tree. For an example of the Expand operator, as shown in Figure 1 (c), since there is no obstacle on the dotted edge $[s_{parent}, s_{new}]$, *i.e.*, $s_{new}$ is reachable, the new state and edge will be added to the search tree (connected by the solid edges).

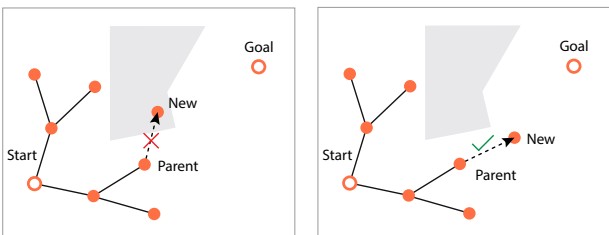

Figure 11: Illustration for one iteration of Algorithm 1. The left and right figures illustrate two different cases where the sample returned by the Expand operator is **unreachable** and **reachable** from the search tree.

Now we will provide two concrete algorithm examples. For instance,

- If we instantiate the Expand operator as Algorithm 4, then we obtain the rapidly-exploring random trees (RRT) algorithm (LaValle, 1998), which first samples a state $s$ from the configuration space

---

**Algorithm 4:** RRT :: Expand$(\mathcal{T}, U)$

---

**Data:** $\mathcal{T} = (\mathcal{V}, \mathcal{E}), U = (s_{init}, \mathcal{S}_{goal}, \mathcal{S}, \mathcal{S}_{free}, \mathtt{map}, c(\cdot))$

1   $s_{rand} \leftarrow \mathcal{U}nif(\mathcal{S})$;                 ▷ Sample configuration space

2   $s_{parent} \leftarrow \mathrm{argmin}_{s \in \mathcal{V}} \|s_{rand} - s\|$;       ▷ Pull to a tree node

3   $s_{new} \leftarrow \mathrm{argmin}_{s \in \mathcal{B}(s_{parent}, \eta)} \|s - s_{rand}\|$;

4   **return** $s_{parent}, s_{new}$;

---

---

**Algorithm 5:** EST :: Expand$(\mathcal{T}, U)$

---

**Data:** $\mathcal{T} = (\mathcal{V}, \mathcal{E}), U = (s_{init}, \mathcal{S}_{goal}, \mathcal{S}, \mathcal{S}_{free}, \mathtt{map}, c(\cdot))$

1   $s_{parent} \sim \phi(s), s \in \mathcal{V}$;               ▷ Sample a tree node

2   $s_{new} \leftarrow \mathcal{U}nif(\mathcal{B}(s_{parent}))$;          ▷ Sample neighborhood

3   **return** $s_{nearest}, s_{new}$;

---

$\mathcal{S}$ and then pulls it toward the neighborhood of current tree $\mathcal{T}$ measured by a ball of radius $\eta$:

$$\mathcal{B}(s, \eta) = \{s' \in \mathcal{S} \mid \|s' - s\| \leqslant \eta\}.$$

Moreover, if the `Postprocess` operator is introduced to modify the maintained search tree as in RRT$^*$ (Karaman & Frazzoli, 2011), the algorithm is provable to obtain the optimal path asymptotically.

- If we instantiate the `Expand` operator as Algorithm 5, then we obtain the expansive-space trees (EST) algorithm (Hsu et al., 1997; Phillips et al., 2004), which samples a state $s$ from the nodes of the existing tree, and then draw a sample from the neighborhood of $s$.

**UCB-based algorithms**    Specifically, in a $K$-armed bandit problem, the UCB algorithm will first play each of the arms once, and then keep track of the average reward $\bar{r}_i$ and the visitation count $n_i$ for each arm. After $T$ rounds of trials, the UCB algorithm will maintain a set of information $\{(\bar{r}_i, n_i)\}_{i=1}^{K}$ with $\sum_{i=1}^{K} n_i = T$. Then, for the next round, the UCB algorithm will select the next arm based on the one-sided confidence interval estimation provided by the Chernoff-Hoeffding bound,

$$a^*_{T+1} = \mathrm{argmax}_{i \in \{1, \ldots, K\}} \, \bar{r}_i + \lambda \sqrt{\frac{\log T}{n_i}}, \tag{7}$$

where $\lambda$ controls the exploration-exploitation trade-off. It has been shown that the UCB algorithm achieves $\mathcal{O}(\log T)$ regret. However, the MCTS is not directly applicable to continuous state-action spaces.

There have been many attempts to generalize the UCB and UCT algorithms to continuous state-action spaces (Chu et al., 2011; Krause & Ong, 2011; Couëtoux et al., 2011; Yee et al., 2016). For instance, contextual bandit algorithms allow continuous arms but involve a non-trivial high dimensional non-convex optimization to select the next arm. In UCT, the progressive widening technique has been designed to deal with continuous actions (Wang et al., 2009). Even with these extensions, the MCTS restricts the exploration only from leaves states, implicitly adding an unnecessary hierarchical structure for path planning, resulting inferior exploration efficiency and extra computation in path planning tasks.

Although these off-the-shelf algorithms are not directly applicable to our path planning setting, their successes show the importance of exploration-exploitation trade-off and will provide the principles for our algorithm for continuous state-action planning problems.

**Planning networks**    Value iteration networks (Tamar et al., 2016) employ neural networks to embed the value iteration algorithm from planning and then use this embedded algorithm to extract input features and define downstream models such as value functions and policies.

Specifically, VIN mimics the following recursive application of Bellman update operator $\mathcal{G}$ to value function $V^*$,

$$V^*(s|U) = (\mathcal{G}V^*)(s) := \min_a \sum_{s'} P(s'|s, a)(c([s, s']) + V^*(s'|U)). \tag{8}$$

where $P(s'|s, a)$ is the state transition model. When the state space for $s$ and action space for $a$ are low dimensional, these spaces can be discretized into grids. Then, the local cost function $c([s, s'])$ and the

value function $V^*(s'|U)$ can be represented as matrices (2d) or tensors (3d) with each entry indexed by grid locations. Furthermore, if the transition model $P(s'|s,a)$ is local, that is $P(s'|s,a) = 0$ for $s' \notin \mathcal{B}(s)$, it resembles a set of convolution kernels, each indexed by a discrete action $a$. And the Bellman update operator essentially convolves $P(s'|s,a)$ with $c([s, s'])$ and $V^*(s'|U)$, and then performs a min-pooling operation across the convolution channels.

Inspired by the above computation pattern of the Bellman operator, value iteration networks design the neural architecture as follows,

$$\tilde{V}^{*0} = \min \left( W_1 \oplus \left[ \texttt{map}, \tilde{R} \right] \right) \tag{9}$$

$$\tilde{V}^{*t} = \min \left( W_1 \oplus \left[ \tilde{V}^{*t-1}, \tilde{R} \right] \right) \tag{10}$$

where $\oplus$ is the convolution operation, both $\texttt{map}$, $\tilde{V}^{*t}$ and $\tilde{R}$ are $d \times d$ matrices, and the parameter $W_1$ are $k_c$ convolution kernels of size $k \times k$. The min implements the pooling across $k_c$ convolution channels.

The gated path planning networks (GPPN) (Lee et al., 2018) improves the VIN by replacing the VIN cell (9) with the well-established LSTM update, *i.e.*,

$$\tilde{V}^t, \tilde{c}^t = \texttt{LSTM} \left( \sum \left( W_1 \oplus \left[ \tilde{V}^t, \tilde{R} \right] \right), \tilde{c}^t \right), \tag{11}$$

where the summation is taking over all the $k_c$ convolution channels.

After constructing the VIN and GPPN, the parameters of the model, *i.e.*, $\left\{ \tilde{R}, W_1 \right\}$ can be learned by imitation learning or reinforcement learning.

The application of planning networks are restricted in low-dimension tasks. However, their success enlightens our neural architecture for generalizable representation for high-dimension planning tasks.

## C  PARAMETRIZED UCB ALGORITHMS

We list two examples of parametrized UCB as the instantiation of (3) used in GPE:

- **GP-UCB**: The GP-UCB Chu et al. (2011) is derived by parameterizing via Gaussian Processes (GP) with kernel $k(s, s')$, *i.e.*, $\mathbb{E}[r(s)|\mathcal{T}, U] \sim \mathcal{GP}(0, k)$, GP-UCB maintains an UCB of the reward after $t$-step as
  $$\phi(s) := \bar{r}_t(s) + \lambda \sigma_t(s), \tag{12}$$
  where
  $$\begin{aligned} \bar{r}_t(s) &= k_t(s)(K_t + \alpha I)^{-1} r_t, \\ \sigma_t^2(s) &= k(s, s) - k_t(s)^\top (K_t + \alpha I)^{-1} k_t(s), \end{aligned}$$
  with $k_t(s) = [k(s_i, s)]_{s_i \in \mathcal{S}_t}$, $K_t = [k(s, s')]_{s, s' \in \mathcal{S}_t}$, and $\mathcal{S}_t = \{s_1, s_2, \dots, s_t\}$ denotes the sequence of selected nodes in current trees. The variance estimation $\sigma_t^2(s)$ takes the number of visits into account in an implicit way: the variance will reduce, as the neighborhood of $s$ is visited more frequently (Srinivas et al., 2009).

- **KS-UCB**: We can also use kernel regression as an alternative parametrization for (7) (Yee et al., 2016), which leads to an UCB of the reward after $t$-step as
  $$\phi(s) := \bar{r}_t(s) + \lambda \sigma_t(s), \tag{13}$$
  where
  $$\begin{aligned} \bar{r}_t[s] &= \frac{\sum_{s' \in \mathcal{S}_t} k(s', s) r(s')}{\sum_{s' \in \mathcal{S}_t} k(s', s)}, \\ \sigma_t(s) &= \sqrt{\frac{\log \sum_{s' \in \mathcal{S}_t} w(s')}{w(s)}}, \end{aligned}$$
  with $w(s) = \sum_{s' \in \mathcal{S}_t} k(s', s)$. Clearly, the variance estimation is to promote exploration towards less frequently visited states.

As we can see, in both two examples of the parametrized UCB, we parametrize the observed rewards, leading to generalizable UCB for increased states by considering the correlations.

# D   POLICY AND VALUE NETWORK ARCHITECTURE

We explain the implementation details of the proposed parametrization for policy and value function. Figure 12 and Figure 13 are neural architectures for the attention module, the policy/value network, and the planning module, respectively.

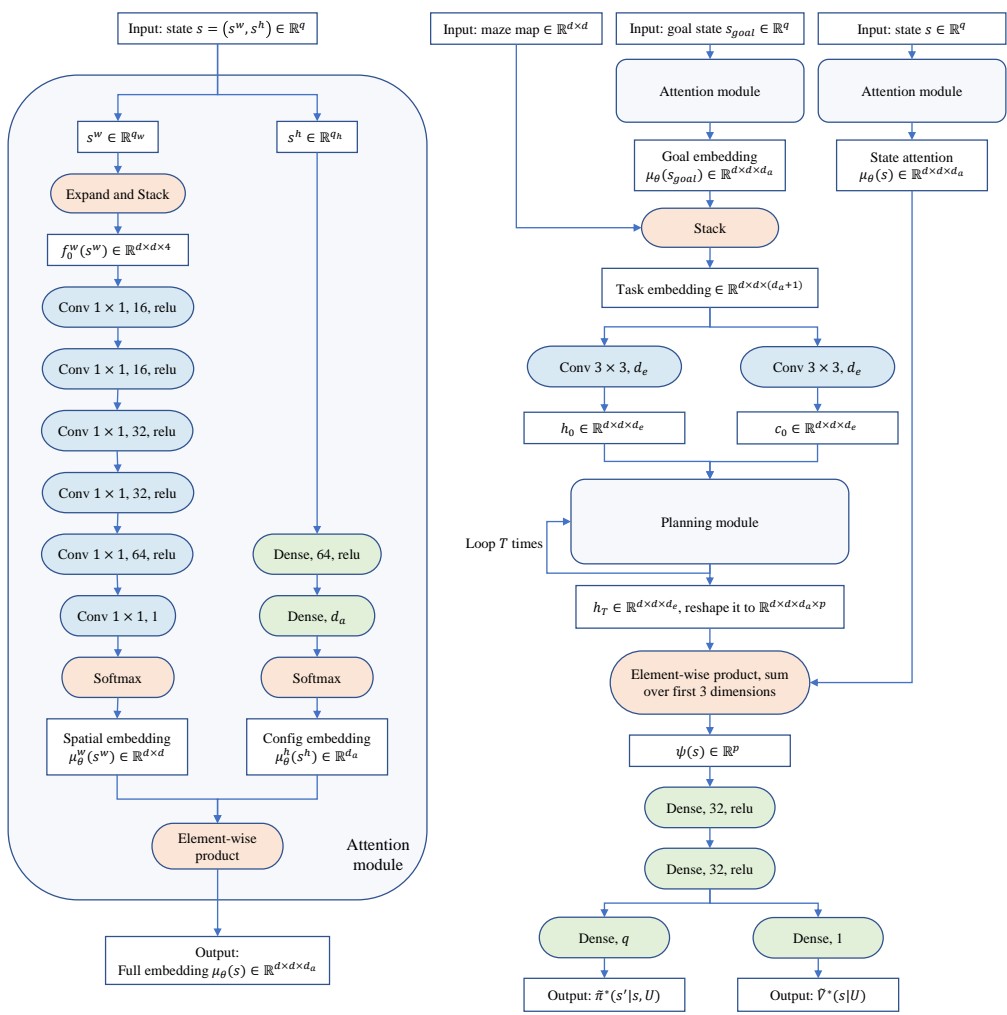

Figure 12: Left: attention module, instantiating the Figure 3; Right: policy/value network, instantiating the Figure 4.

In the figures, we use rectangle blocks to denote inputs, intermediate results and outputs, stadium shape blocks to denote operations, and rounded rectangle blocks to denote modules. We use different colors for different operations. In particular, we use blue for convolutional/LSTM layers, green for dense layers, and orange for anything else. For convolutional layers, "Conv $1 \times 1$, 32, relu" denotes a layer with $1 \times 1$ kernels, 32 channels, followed by a rectified linear unit; for dense layers, "Dense, 64, relu" denotes a layer of size 64, followed by a rectified linear unit.

The attention module (Figure 12-left) embeds a state to a $d \times d \times d_a$ tensor. The planning module (Figure 13) is a one-step LSTM update which takes the result of a convolutional layer as input. Both the input and hidden size of the LSTM cell are $d_e$. All $d \times d$ locations share one set of parameters and are processed by the LSTM in one batch.

The main architecture is illustrated in Figure 12-right. It takes maze map, state and goal as input, and outputs the action and the value. Refer to Section 4.2 for details for computing $\psi(s)$. In our experiments, we set the values of the hyper-parameters to be $(d, d_e, d_a, p) = (15, 64, 8, 8)$.

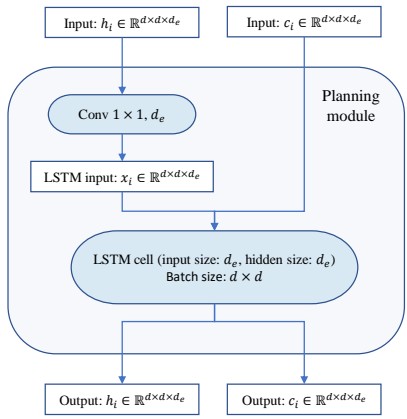

Figure 13: planning module

# E  EXPERIMENT DETAILS

## E.1  BENCHMARK ENVIRONMENTS

We used four benchmark environments in our experiment. For the first three, the workspace dimension is 2d. We generated the maze maps with the recursive backtracker algorithm using the following implementation: https://github.com/lileee/gated-path-planning-networks/blob/master/generate_dataset.py. Examples of the workspace are shown in Figure 15. Three environments differ in the choice of robots:

- **Workspace planning (2d).** The robot is abstracted with a point mass moving in the plane. Without higher dimensions, this problem reduces to planning in the workspace.
- **Rigid body navigation (3d).** A rigid body robot, abstracted as a thin rectangle, is used here. The extra rotation dimension is added to the planning problem. This robot can rotate and move freely without any constraints in the free space.
- **3-link snake (5d).** The robot is a 5 DoF snake with two joints. Two more angle dimensions are added to the planning task. To prevent links from folding, we restrict the angles to the range of $[-\pi/4, \pi/4]$.

The fourth environment has a 3d workspace. Cuboid obstacles were generated uniformly randomly in space with density $\approx 20\%$. Example of the workspace is shown in Figure 6 and 16, where the blue cuboids are obstacles. The environment is described below:

- **Spacecraft planning (7d).** The robot is a spacecraft with a cuboid body and two 2 DoF arms connecting to two opposite sides of the body. There is a joint in the middle of each arm. The outer arm can rotate around this joint. Each arm can also rotate as a whole around its connection point with the body. All rotation angles are restricted in the range of $[0, \pi/2]$. The spacecraft itself cannot rotate.

## E.2  HYPERPARAMETER FOR MSIL

During self-improving over the first 2000 problems, NEXT updated its parameters and annealed $\epsilon$ once for every 200 problems. The value of the annealing $\epsilon$ was set as the following:

$$\epsilon = \begin{cases} 1, & \text{if } i < 1000, \\ 0.5 - 0.1 \cdot \lfloor (i - 1000)/200 \rfloor, & \text{if } 1000 \leqslant i < 2000, \\ 0.1, & \text{otherwise}, \end{cases}$$

with $i$ denoting the problem number.

## E.3  BASELINE: THE IMPROVED GPPN

The original GPPN is not directly applicable to our experiments. Inspired by Tamar et al. (2016), we add a fully-connected MLP to its final layers, so that the improved architecture can be applied to high-dimensional continuous domain. As shown in Figure 14, the GPPN first processes the discretized

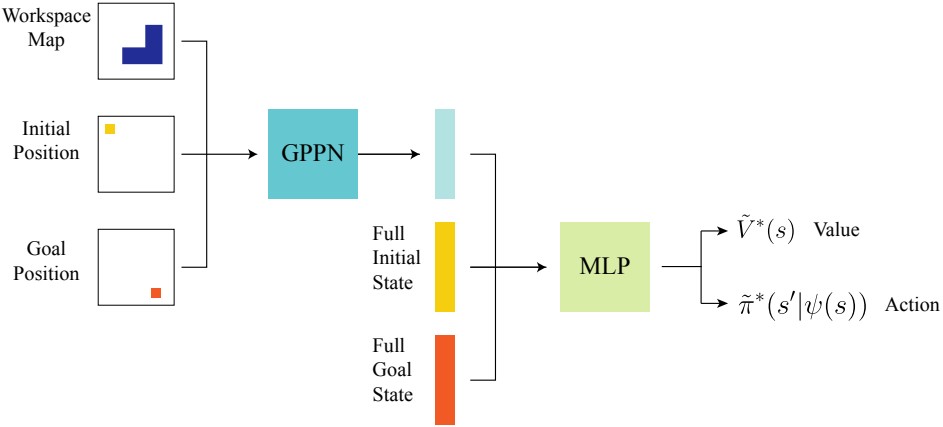

Figure 14: Improved GPPN architecture

workspace locations. Its output and the full robot configurations are processed together by the MLP, which then produces the current value and action estimates. The improved GPPN is trained using supervisions from the near-optimal paths produced by RRT*.

## F    EXPERIMENT RESULTS

### F.1    SOLUTION PATH ILLUSTRATION

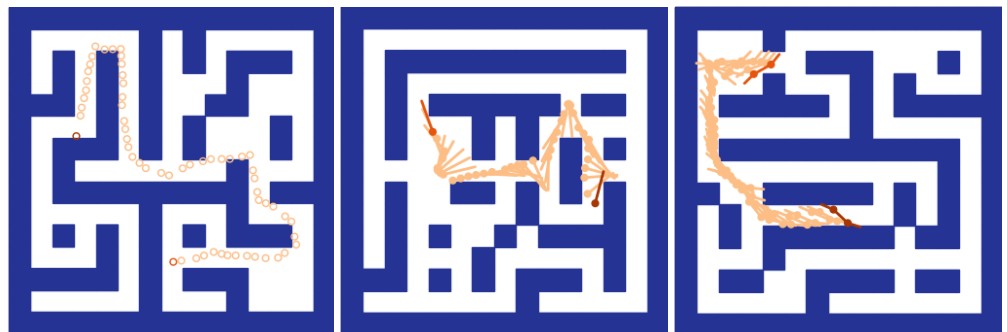

Figure 15:   The solution path produced by NEXT in a workspace planning task (2d), rigid body navigation task (3d), 3-link snake task (5d) from left to right. The orange dot and the brown dot are starting and goal locations, respectively.

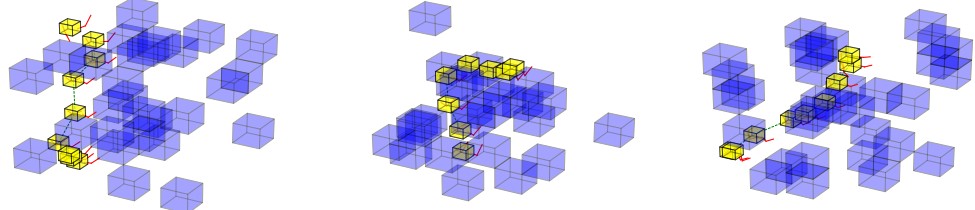

Figure 16:   The solution paths produced by NEXT in spacecraft planning task (7d). Spacecraft has a yellow body and two 2 DoF (red) arms. Blue cuboids are obstacles.

### F.2    DETAILS OF QUANTITATIVE EVALUATION

More detailed results are shown in Table 1, 2, 3, including learning-based and non-learning-based ones, on the last 1000 problems in each experiment. We normalized the number of collision checks and the cost of paths based on the solution of RRT*. The success rate result is not normalized. The

best planners in each experiment are in **bold**. NEXT-KS and NEXT-GP outperform the current state-of-the-art planning algorithm with large margins.

Table 1: Success rate results. The higher the better. NEXT-KS performs the best.

|     | NEXT-KS | NEXT-GP | GPPN-KS | GPPN-GP | RRT* | BIT* | BFS | CVAE | Reject |
|-----|---------|---------|---------|---------|------|------|-----|------|--------|
| 2d | **0.988** | 0.981 | 0.718 | 0.632 | 0.735 | 0.710 | 0.185 | 0.535 | 0.720 |
| 3d | **0.943** | 0.841 | 0.689 | 0.554 | 0.490 | 0.514 | 0.121 | 0.114 | 0.498 |
| 5d | **0.883** | 0.768 | 0.633 | 0.515 | 0.455 | 0.497 | 0.030 | 0.476 | 0.444 |
| 7d | **0.931** | 0.906 | 0.634 | 0.369 | 0.361 | 0.814 | 0.288 | 0.272 | 0.370 |

Table 2: Average number of collision checks results. The lower the better. The score is normalized based on the solution of RRT*. NEXT-KS performs the best in 3 benchmarks.

|     | NEXT-KS | NEXT-GP | GPPN-KS | GPPN-GP | RRT* | BIT* | BFS | CVAE | Reject |
|-----|---------|---------|---------|---------|------|------|-----|------|--------|
| 2d | **0.177** | 0.243 | 2.342 | 3.484 | 1.000 | 5.945 | 9.247 | 1.983 | 1.011 |
| 3d | **0.694** | 1.334 | 2.214 | 3.125 | 1.000 | 7.924 | 7.292 | 2.162 | 0.988 |
| 5d | **0.888** | 1.520 | 1.800 | 2.706 | 1.000 | 7.483 | 5.758 | 1.188 | 0.997 |
| 7d | 0.653 | **0.502** | 1.877 | 1.313 | 1.000 | 4.683 | 3.856 | 1.591 | 0.987 |

Table 3: Average cost of paths. The lower the better. The score is normalized based on the solution of RRT*. The NEXT-KS achieves the best solutions.

|     | NEXT-KS | NEXT-GP | GPPN-KS | GPPN-GP | RRT* | BIT* | BFS | CVAE | Reject |
|-----|---------|---------|---------|---------|------|------|-----|------|--------|
| 2d | **0.172** | 0.193 | 1.049 | 1.333 | 1.000 | 1.140 | 2.811 | 1.649 | 1.050 |
| 3d | **0.116** | 0.315 | 0.612 | 0.875 | 1.000 | 0.955 | 1.720 | 1.734 | 0.984 |
| 5d | **0.215** | 0.426 | 0.673 | 0.890 | 1.000 | 0.923 | 1.780 | 0.961 | 1.020 |
| 7d | **0.108** | 0.147 | 0.573 | 0.987 | 1.000 | 0.291 | 1.114 | 1.139 | 0.986 |

We demonstrated the performance improvement curves for 2d workspace planning, 3d rigid body navigation in Figure 17. As we can see, similar to the performances on 5d 3-link snake planning task in Figure 7, in these tasks, the NEXT-KS and NEXT-GP improve the performances along with more and more experiences collected, justified the self-improvement ability by learning $\tilde{V}^*$ and $\tilde{\pi}^*$.

## F.3 SEARCH TREES COMPARISON

We illustrate the search trees generated by RRT* and the proposed NEXT algorithms with 500 samples in Figure 18, Figure 19, Figure 20 and Figure 21 on several 2d, 3d, 5d and 7d planning tasks, respectively. To help readers better understand how the trees were expanded, we actually visualize the RRT* search trees without edge rewiring, which is equivalent to the RRT search trees, however the vertex set is the same. Comparing to the search trees generated by RRT* side by side, we can clearly see the advantages and the efficiency of the proposed NEXT algorithms. In all the tasks, even in 2d workspace planning task, the RRT* indeed randomly searches without realizing the goals, and thus cannot complete the missions, while the NEXT algorithms search towards the goals with the guidance from $\tilde{V}^*$ and $\tilde{\pi}^*$, therefore, successfully provides high-quality solutions.

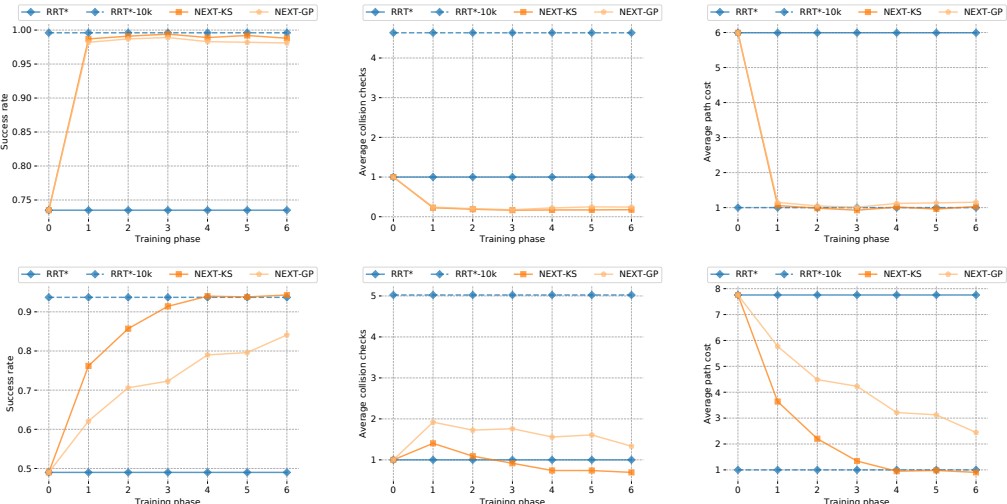

Figure 17: The first and second rows display the improvement curves of our algorithms on all 3000 problems of the 2d workspace planning and 3d rigid body navigation problems. We compare our algorithms with RRT*. Three columns correspond to the success rate, the average collision checks, and the average cost of the solution paths for each algorithm.

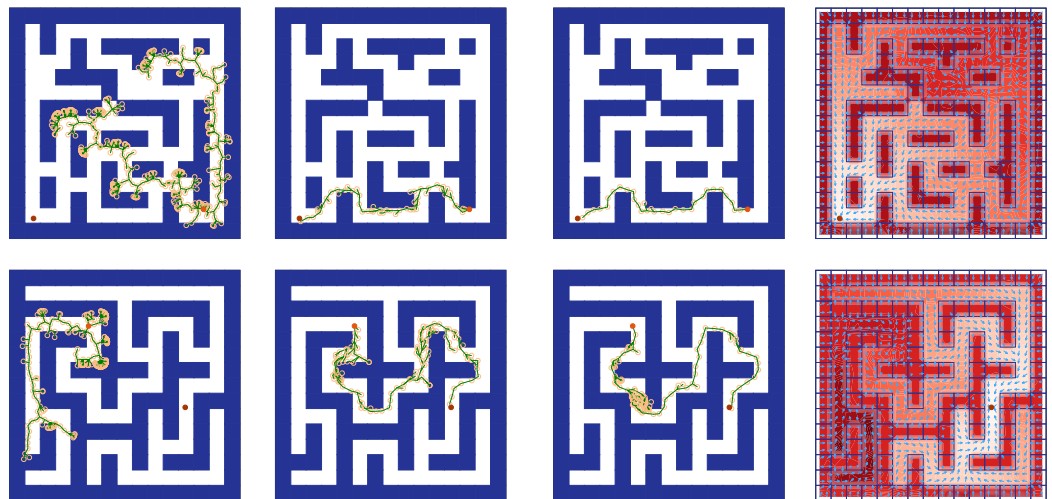

(a) RRT* (w/o rewiring) (b) NEXT-KS search tree (c) NEXT-GP search tree (d) learned $\tilde{V}^*$ and $\tilde{\pi}^*$

Figure 18: Column (a) to (c) are the search trees produced by the RRT*, NEXT-KS, and NEXT-GP on the same workspace planning task (2d). The learned $\tilde{V}^*$ and $\tilde{\pi}^*$ from NEXT-KS are plotted in column (d). In the figures, obstacles are colored in deep blue, the starting and goal locations are denoted by orange and brown dots, respectively. In column (a) to (c), samples are represented with hollow yellow circles, and edges are colored in green. In column (d), the level of redness denotes the value of the cost-to-go estimate $\tilde{V}^*$, and the cyan arrows point from a given state $s$ to the center of the proposal distribution $\tilde{\pi}^*(s'|s, U)$. We set the maximum number of samples to be 500.

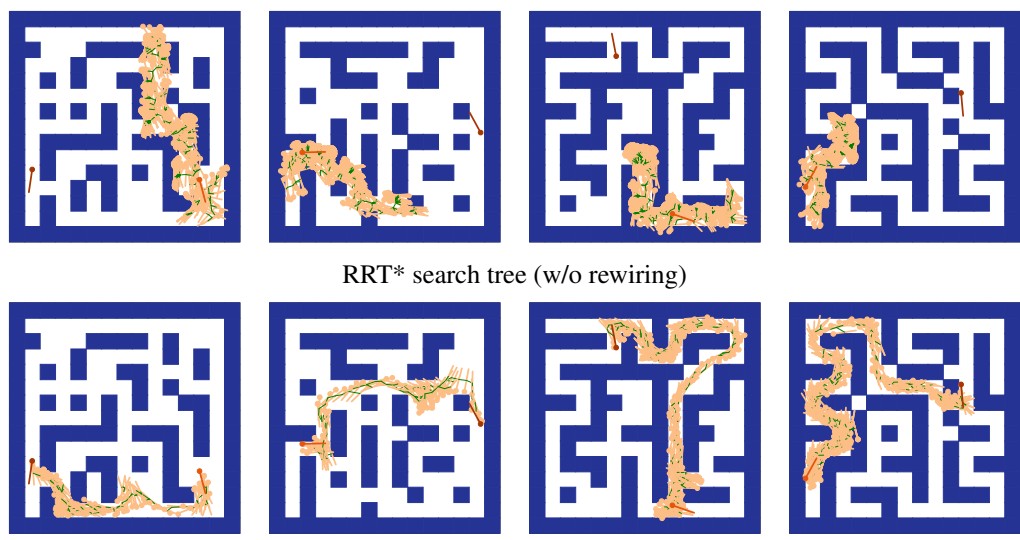

RRT* search tree (w/o rewiring)

NEXT-KS search trees

Figure 19: Each column corresponds to one example from the rigid body navigation problem (3d). The top and the bottom rows are the search trees produced by the RRT* and NEXT-KS, respectively. In the figures, obstacles are colored in deep blue, and the rigid bodies are represented with matchsticks. The samples, starting states, and goal states are denoted by yellow, orange, and brown matchsticks, respectively. Edges are colored in green. We set the maximum number of samples to be 500.

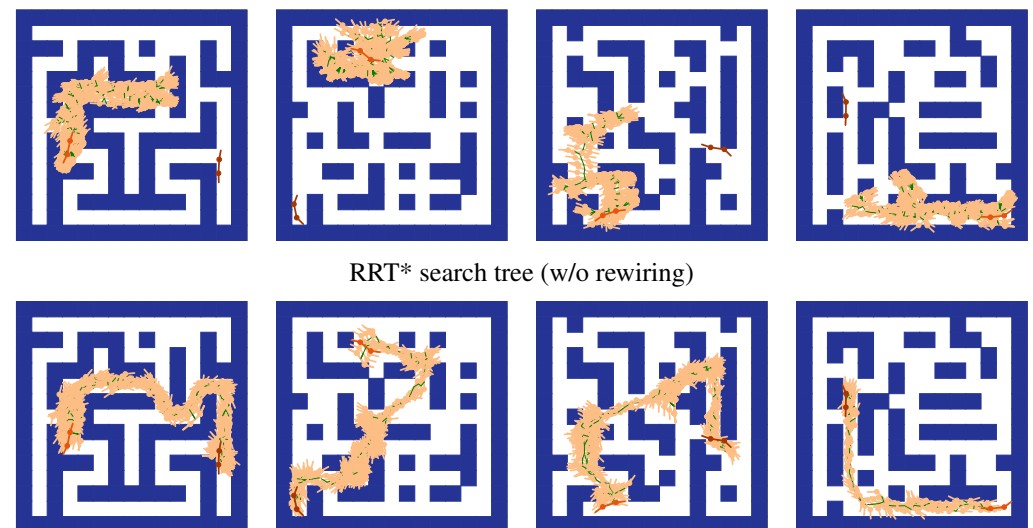

RRT* search tree (w/o rewiring)

NEXT-KS search trees

Figure 20: Each column corresponds to one example from the 3-link snake problem (5d). The top and the bottom rows are the search trees produced by the RRT* and NEXT-KS, respectively. In the figures, obstacles are colored in deep blue, and the rigid bodies are represented with matchsticks. The samples, starting states, and goal states are denoted by yellow, orange, and brown matchsticks, respectively. Edges are colored in green. We set the maximum number of samples to be 500.

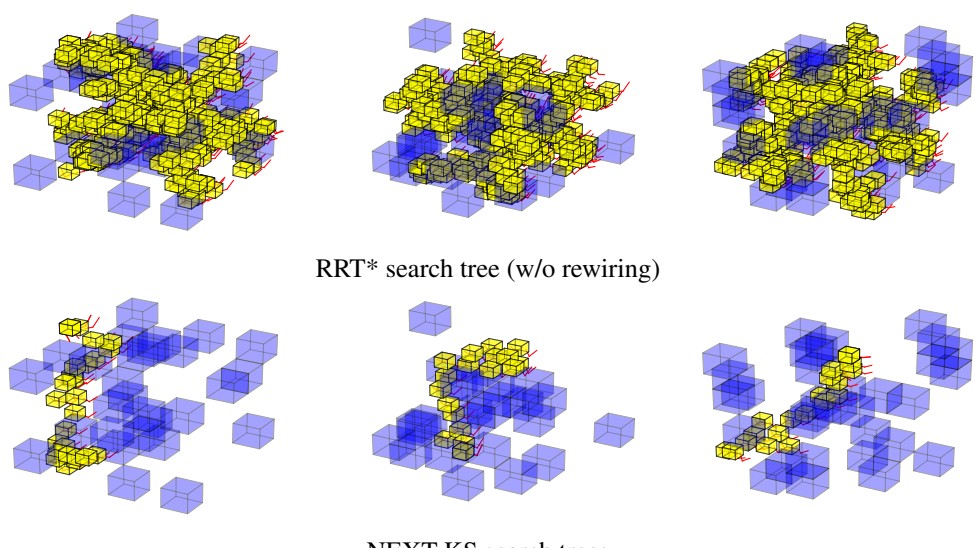

RRT* search tree (w/o rewiring)

NEXT-KS search trees

Figure 21: Each column corresponds to one example from the spacecraft planning problem (7d). The top and the bottom rows are the search trees produced by the RRT* and NEXT-KS, respectively. In the figures, obstacles are colored in blue, and each spacecraft has a yellow body and two 2 DoF red arms. We set the maximum number of samples to be 500.

### F.4 CASE STUDY DETAILS

We conduct a real-world case study on controlling robot arms to move objects on a shelf. This is a representative of common scenarios in practice where the robot needs to plan its motion in real-time to reach the inside of some narrow space. For this case study, we focus more on the practical aspect to evaluate how much can we improve on the wall-clock time by learning from similar planning problems.

### F.4.1 TASK DESCRIPTION

We generate planning problems randomly to form the training set and test set. In each planning task, there is a shelf of multiple levels, with each level horizontally divided into multiple bins. The heights of levels and widths of bins are randomly drawn from some fixed distribution. Samples of shelves are shown in Figure 22. Both the start and goal configurations are randomly sampled from a distribution within the reachable region of the robot arm. The planning environment is created with the OpenRave simulator (Diankov, 2010).

The task is to find a path for the 7 DoF robot arm to move from a location in one bin to another, *i.e.*, the end effectors of the start and goal configurations are in different bins, as illustrated in Figure 23. In this case, the base of the robot is fixed and we are planning the movement of arm. We generated 3000 problems for training and 1000 problems for testing.

### F.4.2 BASELINES AND TRAINING

For traditional planners, we include `C++` OMPL (Şucan et al., 2012) implementation of BIT* (Gammell et al., 2015) and RRT* (Karaman & Frazzoli, 2011) as baselines. The hyperparameters of RRT* and BIT* are specially tuned for this experiment. We also compare with learning-based planners CVAE-plan (Ichter et al., 2018) and Reinforce-plan (Zhang et al., 2018). The supervisions for CVAE-plan are produced by the well-tuned BIT* on the training set. To train NEXT, we consider the BIT* instead of RRT* as the imperfect expert in training problems.

### F.4.3 RESULTS

We evaluate the algorithms on the separated testing problems, and record the success rate using 10 different time limits. The success rate and average path quality are plot in Figure 8 and recorded in Table 4 and Table 5. The solution paths found by NEXT and BIT* are illustrated in Figure 23. In terms of both success rate and solution path quality, NEXT dominates all the planners under all time limits.

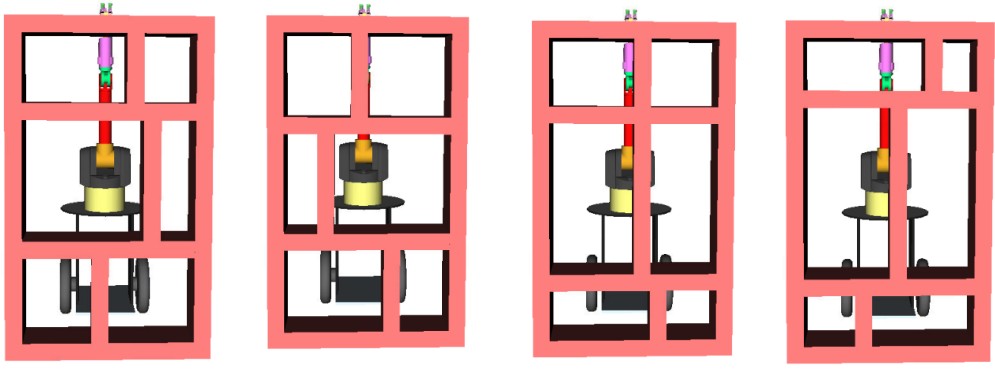

Figure 22: Examples of different shelves sampled from the distribution.

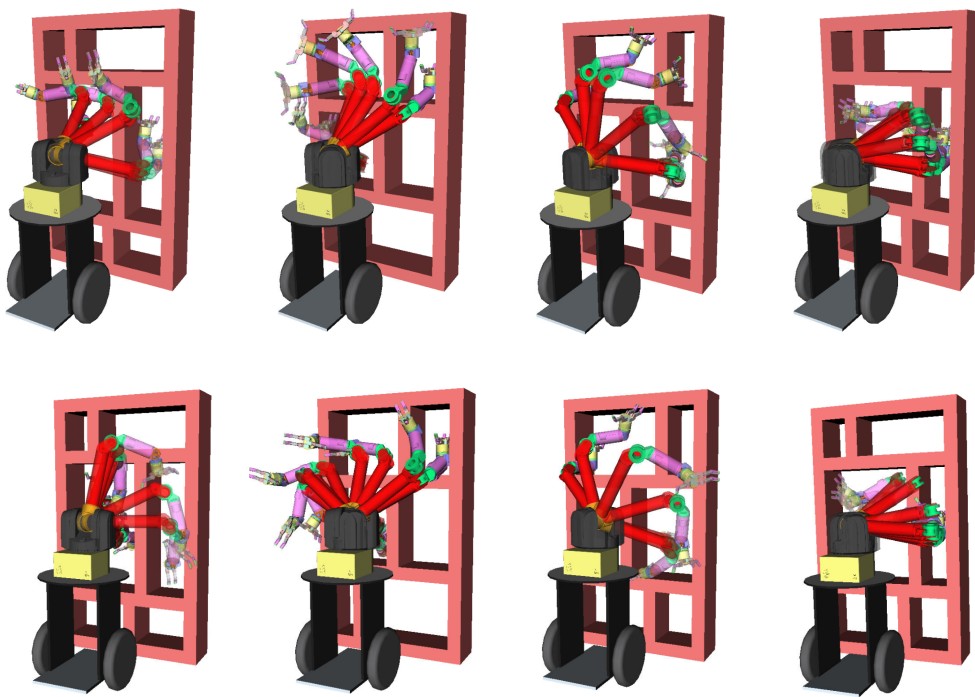

Figure 23: First row: robot arm solution trajectories produced by NEXT(-KS) in four planning problems; Second row: BIT* solutions on the same planning problems. NEXT only takes 5 seconds to complete each problem while BIT* needs 250 seconds to find a solution for the hardest problems (last two columns).

Table 4: Success rates of different planners under varying time limits, the higher the better.

|  | 5s | 10s | 15s | 20s | 25s | 30s | 35s | 40s | 45s | 50s |
|---|---|---|---|---|---|---|---|---|---|---|
| NEXT | **0.579** | **0.657** | **0.703** | **0.709** | **0.745** | **0.743** | **0.746** | **0.752** | **0.772** | **0.763** |
| CVAE | 0.354 | 0.437 | 0.482 | 0.509 | 0.507 | 0.539 | 0.553 | 0.551 | 0.579 | 0.580 |
| Reinforce | 0.160 | 0.170 | 0.200 | 0.150 | 0.180 | 0.190 | 0.175 | 0.180 | 0.225 | 0.175 |
| BIT* | 0.226 | 0.288 | 0.320 | 0.365 | 0.364 | 0.429 | 0.425 | 0.422 | 0.443 | 0.475 |
| RRT* | 0.135 | 0.148 | 0.144 | 0.136 | 0.147 | 0.159 | 0.152 | 0.158 | 0.157 | 0.165 |

Table 5: Average path costs of different planners under varying time limits, the lower the better.

|  | 5s | 10s | 15s | 20s | 25s | 30s | 35s | 40s | 45s | 50s |
|---|---|---|---|---|---|---|---|---|---|---|
| NEXT | **25.143** | **21.873** | **20.052** | **19.773** | **18.318** | **18.381** | **18.274** | **18.047** | **17.224** | **17.587** |
| CVAE | 34.414 | 30.815 | 28.909 | 27.717 | 27.784 | 26.408 | 25.776 | 25.883 | 24.665 | 24.659 |
| Reinforce | 42.781 | 42.471 | 41.142 | 43.295 | 42.129 | 41.657 | 42.235 | 42.093 | 40.269 | 42.282 |
| BIT* | 39.543 | 37.086 | 35.744 | 34.002 | 34.007 | 31.414 | 31.587 | 31.651 | 30.841 | 29.601 |
| RRT* | 43.368 | 42.855 | 42.981 | 43.355 | 42.919 | 42.410 | 42.680 | 42.441 | 42.504 | 42.165 |

