# OpenReview forum: "Learning to Plan in High Dimensions via Neural Exploration-Exploitation Trees"
_ICLR.cc/2020/Conference — Accept (Spotlight)_

### Official Review · AnonReviewer2 · 2019-10-23
**Official Blind Review #2**

**Rating:** 6

**Review:**

This paper proposes an approach to learn how to plan in continuous spaces using neural
nets to learn a value function and a policy for scoring and sampling next-step candidates
in a stochastic tree search.  The networks are updated as more planning tasks
are executed, producing more data for the policy and value function, leading to gradually
better plans compared to a number of baselines on benchmarks introduced by the authors.

This is a very interesting paper, although I did not always found it easy to read,
maybe too densely packed for comfort. My main concerns are clarity of the exposition (especially
of the neural net architecture (sec 4.2) and that the comparisons are exclusively done
on benchmarks introduced by the authors rather than on benchmarks on which the baseline
methods had been previously been optimized, which may introduce a bias in favour of the
proposed approach.

Clarifications

Before eqn 2, I don't understand why U includes both S_free and map, although the map specifies the free space and thus S_free seems redundant.

In sec 3 (page 3), the authors introduce a new notation s_init which seems to be the same as s_0 in the previous sections (or is it?).

Section 4.2 was really difficult for me to parse and is too compressed (so is the rest of the paper but this one was worse).

Figures were too small (esp. fig 4 and fig 7) for me to read from the printed paper.

The term 'meta self-improving learning' seems inappropriate. I did not see  how this was a form of meta-learning. Unless I missed something I suggest to change the terminology.

Other Concerns

I have a concern regarding the way r_t(s) is estimated (page 4) by kernel interpolation of the rewards. I fear that it will not generalize properly when trying to extrapolate, especially in high dimensions (since the claim of the paper is that the proposed algorithms is meant for 'high dimensional' states).

In addition, the experiments are actually performed in  rather low-dimensional settings (compared to working on problems with perceptual inputs, for example).



**Experience Assessment:**

I have read many papers in this area.

**Review Assessment: Checking Correctness Of Derivations And Theory:**

N/A

**Review Assessment: Checking Correctness Of Experiments:**

I assessed the sensibility of the experiments.

**Review Assessment: Thoroughness In Paper Reading:**

I read the paper at least twice and used my best judgement in assessing the paper.

---

> ### Author Response · Authors · 2019-11-15
> **Response to Reviewer #2**
>
> Thanks for the generally positive and constructive comments. We have refined paper accordingly to make the paper less intensive.
>
> We address the raised concerns below:
>
> - Experimental environment.
>
> For planning tasks, we did a thorough search and could not find one benchmark which (1) is used by many existing works, (2) is publicly available, and (3) has Python interface.
>
> We are aware of the importance of making fair comparisons with existing methods. Therefore we created one benchmark, based on existing papers [Tamar et al. 2016, Lee et al. 2018, Ichter et al. 2018], without any inductive bias towards the proposed planner. Moreover, we carefully tuned the hyperparameters for all baselines. We would be grateful if you could point us to some benchmarks that meet the criteria.
>
> In the updated version we include a new set of comparisons on a new environment, in which we show our algorithm also performs well when controlling a 7-DOF robot arm to fetch objects from a shelf. Similar (and much easier) experiments are done in BIT* [Gammell et al. 2015]. We show that our algorithm only requires 1/50 wall-clock time compared with the well-tuned BIT* and RRT*.
>
> We will release the code and the environment platform to prosper future research in this direction.
>
> - Notation and terminology clarification.
>
> $\mathtt{map}$ and $\mathcal{S}_{free}$: The $\mathtt{map}$ and $\mathcal{S}_{free}$ are only redundant in 2D case. For a high-dimensional planning problem, the map only captures the workspace information, while the $S_{free}$ will contain the feasible configurations of the robot, e.g., the states of each joint and arm of the robot. While it is easy to operate on $\mathtt{map}$, it would be difficult to work with $\mathcal{S}_{free}$ directly.  In general, for sampling-based path planners, the only way to access $\mathcal{S}_{free}$ is through calling the collision detection oracle.
>
> $s_0$ and $s_{init}$: They are different. The $s_{init}$ stands for the required starting point in the planning task, while $s_0$ stands for the starting point of one path. The planning algorithm is looking for a feasible path whose $s_0 = s_{init}$ and $s_T \in \mathcal{S}_{goal}$.
>
> Meta self-improving learning: As we discussed, each individual planning problem itself is an optimization problem (i.e., to minimize the path cost). We are learning an algorithmic planning solver, which can be generalized for solving other unseen planning problems in the same task distributions.  Therefore the process can be viewed as a meta-learning procedure.
>
>
> - Regarding the 'high-dimensional' planning tasks.
>
> Thanks for the suggestion. It should be emphasized that the dimension in planning tasks usually refers to the controllable variables in configuration space. This is different from the problems with perceptual inputs, where the dimension refers to the observation. Actually, the research on perceptual inputs is a separate topic that is orthogonal to the problem the proposed NEXT targeting, and can be incorporated with NEXT to handle complicated agent planning with observed perceptual contexts.
>
> With the clarification of the 'high-dimension', compared with the current learning to plan algorithms, e.g., VIN and GPPN which are restricted to handle a particle in 2D discrete domains, the proposed NEXT is naturally suitable for complicated agents with more controllable variables.
>
> The proposed NEXT is compatible with arbitrary contexture bandit algorithms. We can definitely incorporate with learned neural networks as features to conduct the UCB estimation, which itself is of independent interest. In our paper, we exploited and tested the major dominated contexture UCB algorithms based on kernels, i.e., GP-UCB and KS-UCB. In practice, the kernel method performs comparably with neural networks when the number of dimensions is hundreds, as shown in [Dai et al. 2014], which is already considered high-dimensional planning tasks.
>
> [Tamar et al. 2016] Tamar, A., Wu, Y., Thomas, G., Levine, S., and Abbeel, P. Value iteration networks. In Advances in Neural Information Processing Systems, pp. 2154–2162, 2016.
> [Lee et al. 2018] Lee, L., Parisotto, E., Chaplot, D. S., Xing, E., and Salakhutdinov, R. Gated path planning networks, 2018.
> [Ichter et al. 2018] Ichter, B., Harrison, J., and Pavone, M. Learning sampling distributions for robot motion planning. In 2018 IEEE International Conference on Robotics and Automation (ICRA). 2018.
> [Gammell et al. 2015] Gammell, J. D., Srinivasa, S. S., and Barfoot, T. D. Batch informed trees (bit*): Sampling-based optimal planning via the heuristically guided search of implicit random geometric graphs. In Robotics and Automation (ICRA), 2015 IEEE International Conference on, pp. 3067–3074. IEEE, 2015.
> [Dai et al. 2014] Dai, B., Xie, B., He, N., Liang, Y., Raj, A., Balcan, M. F. F., & Song, L. (2014). Scalable kernel methods via doubly stochastic gradients. In Advances in Neural Information Processing Systems (pp. 3041-3049).

---

### Official Review · AnonReviewer1 · 2019-10-24
**Official Blind Review #1**

**Rating:** 8

**Review:**

Summary:

Motion-planning in high dimensional spaces is challenging due to the curse of dimensionality. Sampling-based motion planners like PRM, PRM*, RRT, RRT*, BIT* etc have been the go-to solution family. But often these algorithms solve every planning problem tabula rasa. This work combines learning with sampling-based planning such that the parent-sampling and expansion steps instead of being done by common heuristics are learnt in an online manner. Also the resulting exploration-exploitation problem is naturally dealt via using a UCB-style contextual bandit algorithm. Since the number of parents are always varying the common trick of 'describe the action choices with respect to the environment' is adopted so that varying number of actions (states to be sampled from) can be naturally incorporated.

The other significant aspect of this paper is that there is a self-improving component (Algorithm 3) where a dataset is built up every time step, of environments where either an expansion with RRT or the learnt expansion policy is attempted with the policy being invoked more as time goes on and it trains more. If the process succeeds in finding a path to the goal then this example is added to a dataset and the dataset used to update the policy and associated value function to guide it towards the feasible paths found in the tree.

Comments:


- Algorithm 3: "Reconstruct optimal path". These paths are not really optimal for the problem. They are optimal in the tree T that is built so far for example U. But for the problem they are feasible and if RRT* were to be run asymptotically then perhaps near-optimal. The accompanying text should be updated accordingly so that there isn't confusion.

- Here is my main concern with Algorithm 3: For equation 6  where the policy and value functions are updated, the policy is inevitably going to suffer from covariate shift. This is because the algorithm is essentially doing behavior cloning (BC) with respect to the feasible paths found on the planning examples. Since we are inherently in a sequential setting (non-iid) where the states visited by the policy are a direct result of its own decisions the error bound will be quadratic in the horizon (path-length) for equation 6. This phenomenon has been well-understood in imitation learning literature and algorithms like DAgger, AggreVate or online versions like AggreVateD, LOLS already address these problems in a principled manner. Equation 6 should ideally be replaced with an inner DAgger/AggreVateD like loop (with an RRT* dynamic oracle) for stable learning of policy and value function. I am happy to be convinced that covariate shift and resulting quadratic mistake-bound problems are not present here.

- Application of imitation learning to both self-improvement style path planning and leveraging experience in planning has been done before: See "Learning to Search via Retrospective Imitation
Jialin Song, Ravi Lanka, Albert Zhao, Aadyot Bhatnagar, Yisong Yue, Masahiro Ono, 2018" (this is unpublished it seems so it is unfair of me to mention this perhaps but I wanted to give an example of how to use dynamic oracles for stable imitation in planning.) and "Data-driven Planning via Imitation Learning
Sanjiban Choudhury, Mohak Bhardwaj, Sankalp Arora, Ashish Kapoor†, Gireeja Ranade, Sebastian Scherer and Debadeepta Dey", IJRR 2018. At least the last paper should be cited and discussed in related work.

- Also would be curious how the authors would situate methods which are non-learning based but leverage experience in planning (example E-Graphs: Bootstrapping Planning with Experience Graphs, Phillips et al, RSS 2012) via graphs discovered in other problems directly. Perhaps a discussion in related work is warranted?

Update: After rebuttal updating to Accept.


**Experience Assessment:**

I have published in this field for several years.

**Review Assessment: Checking Correctness Of Derivations And Theory:**

N/A

**Review Assessment: Checking Correctness Of Experiments:**

I assessed the sensibility of the experiments.

**Review Assessment: Thoroughness In Paper Reading:**

I read the paper thoroughly.

---

> ### Author Response · Authors · 2019-11-15
> **Response to Reviewer #1**
>
> Thanks for the generally positive comments and constructive suggestions. We address the corresponding concerns below:
>
> 1, Covariate shift and quadratic error
>
> There might be several misunderstandings about our learning setting and the proposed meta self-improving learning (MSIL) algorithm. The major difference between our learning setting and traditional imitation learning, which is used in the existing VIN and GPPN, is that we do not have the optimal expert supervision. Therefore, neither vanilla Behavior Cloning (BC) and the advanced imitation learning algorithms, e.g., DAgger/AggreVateD, can be straightforwardly applied to achieve better performance.
>
> We inherit the design philosophy in DAgger, i.e., "correcting the decisions via guidance from the experts upon the trajectories obtained by the mixture of experts and current imitator", which lead to the MSIL as illustrated in Algorithm 3. In fact, the proposed MSIL can be viewed as a variant of DAgger with the RRT* as the ‘imperfect’ expert. The data collection is executing a mixture of RRT* and current imitator in line 4 in Algorithm 3, which is different from BC where the samples purely come from the imperfect RRT*. Therefore, the covariate shift and quadratic error should not be presented.
>
>
> 2, Related work
> Thanks for the pointer to the references. We have added the discussions into the update version.
>
> - Comparison to [Song et al. 2018]
> The [Song et al., 2018] indeed shares some similarities in the learning part of NEXT, in the sense that both are handling imperfect experts. However, there are significant differences as our major contributions:
> 1, The planning tree expansion procedure is different. In [Song et al., 2018], the next search step must be conducted from the current node, while in our NEXT, the search tree expands from an arbitrary node, saving a huge cost for sampling.
> 2, To compensate for the sub-optimality of the current solution, we exploit the UCB algorithm to balance exploration and exploitation.
> 3, We propose a novel neural network architecture to embed the planning tasks, on which the learned planner can be generalized for future tasks.
>
> As shown in our ablation study, all these components are important to achieve the superb performances.
>
> - Comparison to [Choudhury et al., 2018]
> In the previous version, we already cited and discussed the conference version of this paper [Bhardwaj et al., 2017]. We have changed it to the comprehensive journal version. The paper learns a policy to do search-based planning. However, their method is restricted to planning on graphs.
>
> - Discussion about [Phillips et al., 2012]
> In NEXT, we are targeting a different problem setting compared with non-learning, memory-based planners such as PRM and E-Graphs. The latter planners are designed for largely fixed obstacles, so that one can leverage the search graphs created in previous planning problems to accelerate planning. Our method instead only assumes the planning problems follow some distribution, so that the learned planner can be generalized to unseen tasks from the same distribution.
> 3, Other Questions
>
> - “Reconstruct optimal path” in Algorithm 3
> We have changed it to “nearly-optimal/sub-optimal path” in the updated version.
>
>
> [Bhardwaj et al., 2017] Bhardwaj, Mohak, Sanjiban Choudhury, and Sebastian Scherer. "Learning heuristic search via imitation." arXiv preprint arXiv:1707.03034 (2017).
> [Choudhury et al., 2018] Choudhury, Sanjiban, et al. "Data-driven planning via imitation learning." The International Journal of Robotics Research 37.13-14 (2018): 1632-1672.
> [Phillips et al., 2012] Phillips, Mike, et al. "E-Graphs: Bootstrapping Planning with Experience Graphs." Robotics: Science and Systems. Vol. 5. No. 1. 2012.

---

### Official Review · AnonReviewer3 · 2019-10-27
**Official Blind Review #3**

**Rating:** 8

**Review:**

The paper introduces a novel  meta path planning algorithm  that utilizes neural network module that improves the data-efficiency for iterated path planning problems.

The authors address a relevant issue and the experiments make sense given the research question.  I particular like the 3 ablation studies that the authors include, which makes the empirical analysis very thorough.

Writing and Clarity:
The introduction is written quite well. Section II&III is written quite technical and dense. This can be very hard to understand for non-experts. However these section are  important to understand the  rest paper. Finally, these two sections should be integrated (preliminaries, quite literally, should be at the beginning). Section 5


Additional Questions:
1. Philosophically, how does the self-improvement for iterative planning problems not contradict the no-free lunch theorem? What kind of repeated structure do we assume here (because it seems as in Fig. 1 both the obstacles as well as the goal state change randomly)
2. As you employ a neural network to do value iteration how does the wall-clock time compare to the baselines?  I do not mean the environment time-ticks (that you checked for using the number of collision checks), but actual compute time.
3. How sensitive is the proposed solution to parameter initialization?  Did you find much variation in changing hyper-parameters, such as network topology, learning rate et cetera?


**Experience Assessment:**

I do not know much about this area.

**Review Assessment: Checking Correctness Of Derivations And Theory:**

I assessed the sensibility of the derivations and theory.

**Review Assessment: Checking Correctness Of Experiments:**

I assessed the sensibility of the experiments.

**Review Assessment: Thoroughness In Paper Reading:**

I read the paper at least twice and used my best judgement in assessing the paper.

---

> ### Author Response · Authors · 2019-11-15
> **Response to Reviewer #3**
>
> Thanks for your support and inspiring comments. We reply to the raised questions below:
>
> 1. Neural induced prior in "no-free lunch theorem".
>
> We are not seeking the omniscient planner dominating any other alternatives, which is impossible based on the no-free lunch theorems for optimization. Our underlying assumption, as discussed in Section 2, is that the learned planner will do better, in terms of collision check and quality, for planning problems from certain task distribution.
>
> As an example, for the distribution that only generates planning problems in which $\mathcal{S}_{goal}$ always lies in the 1 step right of $s_{init}$ in the mazes without any obstacles, the trivial planner that only produces 1 step right action will be the optimal planner. Although the learned planner pays for the degraded performances on this set of trivial problems, it will take the advantages in the neural network encoded repeated patterns in the practical tasks, which we are indeed interested in.
>
>
> 2. Computing wall-clock comparison.
>
> We used the number of collision checks and the number of samples as a surrogate for time because collision check for each sample is time-consuming and in many cases it dominates the total planning time. We have shown that asymptotically our method is more efficient than the existing ones.
>
> As we show in the newly added experiments in Figure 8, to achieve the same success rate, our algorithm implemented in Python only requires 1/50 wall-clock time compared to the highly-optimized OMPL implementation in C of traditional planners such as BIT* and RRT*. The result shows a huge potential to improve the actual planning time in real-world tasks using our algorithm.
>
> In each iteration, the algorithm calls the network to generate the next sample. We would like to emphasize that there is a computation trick to save the major compute time. As we discussed in the main text, the latent representation of the value/policy function $\nu^{*(T)}$ defined before equation (5) is independent of the current state $s$. Therefore we can precompute $\nu^{*(T)}$ and reuse that in each iteration, which avoids duplicating the efforts doing the neuralized value iteration. Using this trick the algorithm runs 10x - 100x faster than before.
>
> 3. Parameter/network component sensitivity.
>
> We did not find large variations in changing the hyper-parameters and network component:
>     - For neural network weights and biases, we used the default initialization (kaiming_uniform_) for nn layers (Linear/Conv) in Pytorch 1.0. Switching it to kaiming_normal_, xavier_uniform_, or xavier_normal_ does not improve/degrade the result by more than 1%.
>     - For optimization, we used Adam optimizer in Pytorch. We set betas to (0.9, 0.999) but did not tune it. The learning rates <= 1e-3 works fine.
>     - For latent dimensions in our neural network, if we vary each hyperparameter in the range of [0.5*r, 2*r], where r is the recommended value (in Appendix D), the variation is within 5%.

---

### Decision · Program_Chairs · 2019-12-19

**Decision:**

Accept (Spotlight)

**Comment:**

All reviewers unanimously accept the paper.